# A Closer Look at the Robustness of Contrastive Language-Image Pre-Training (CLIP)

**Weijie Tu**[1]    **Weijian Deng**[1]    **Tom Gedeon**[2,3]
[1]The Australian National University    [2]Curtin University [3]University of ÓBuda
firstname.lastname@anu.edu.au    firstname.lastname@curtin.edu.au

## Abstract

Contrastive Language-Image Pre-training (CLIP) models have demonstrated remarkable generalization capabilities across multiple challenging distribution shifts. However, there is still much to be explored in terms of their robustness to the variations of specific visual factors. In real-world applications, reliable and safe systems must consider other safety objectives beyond classification accuracy, such as predictive uncertainty. Yet, the effectiveness of CLIP models on such safety-related features is less-explored. Driven by the above, this work comprehensively investigates the safety objectives of CLIP models, specifically focusing on three key properties: resilience to visual factor variations, calibrated uncertainty estimations, and the ability to detect anomalous inputs. To this end, we study 83 CLIP models and 127 ImageNet classifiers. They are diverse in architecture, (pre)training distribution and training strategies. We consider 10 visual factors (*e.g.*, shape and pattern), 5 types of out-of-distribution data, and 8 natural and challenging test conditions with different shift types, such as texture, style, and perturbation shifts. Our study has unveiled several previously unknown insights into CLIP models. For instance, they are not consistently more calibrated than other ImageNet models, which contradicts existing findings. Additionally, our analysis underscores the significance of training source design by showcasing its profound influence on the three safety-related properties. We believe our comprehensive study can shed light on and help guide the development of more robust and reliable CLIP models.

## 1   Introduction

By leveraging natural language supervision, CLIP has made significant progress in enhancing the zero-shot capabilities of models, unleashing their potential for remarkable out-of-distribution generalization performance [1, 2]. For example, CLIP models perform zero-shot classification without being explicitly trained on the target dataset, and they exhibit strong robustness to challenging natural distributional shifts [3–7]. Understanding such behavior of CLIP models is critical for advancing the next generation of image-text foundation models. Current research on this topic has explored various aspects of CLIP models, including dataset creation [8], reproducible scaling law [9], fine-tuning approaches [10], and training distribution [11].

In this work, we conduct an in-depth analysis of the safety-related properties of CLIP models, with a particular emphasis on their robustness and reliability across diverse testing environments. Specifically, we delve into the three critical safety-related properties, namely: 1) robustness to visual factors, to assess whether CLIP models can maintain robust when encountering varying visual factors, such as pose, size, color, lighting, and occlusions; 2) out-of-distribution (OOD) detection, to evaluate the capability of CLIP models to detect instances with labels that are not part of the training distribution; and 3) predictive uncertainty, to investigate whether CLIP models can provide calibrated predictions that accurately reflect their uncertainty in different testing conditions.

37th Conference on Neural Information Processing Systems (NeurIPS 2023).

Our research offers a comprehensive examination of advantages and drawbacks of CLIP models across various critical facets. Building on prior research that has highlighted robustness of CLIP, we further study their performance across specific factors such as pose and lighting. Additionally, although CLIP models have demonstrated their efficacy in detecting OOD data [12], it is still uncertain whether this ability remains consistent when the training distribution, fine-tuning datasets, model size, and architecture are altered. Moreover, beyond classification accuracy, it is also important to evaluate whether CLIP models offer reliable uncertainty estimation across various distributions.

In light of the aforementioned questions, we evaluate 51 zero-shot CLIP models with varying visual encoder architectures, training sources, and dataset sizes, as well as 32 ImageNet fine-tuned CLIP models. To establish a baseline, we compare these models against 127 ImageNet models without language-image pre-training. We examine 10 visual factors variations present in the ImageNet validation set [13], including object pose, lighting, and background, to assess models' visual factors-level robustness. As for OOD detection, we employ ImageNet as an in-distribution (ID) set following [12] and test on 5 types of OOD scenarios. Furthermore, to investigate the predictive uncertainty, we use a set of canonical ImageNet distributions, such as texture, style, and perturbation shifts. Below we present key observations and insights obtained from our study:

- CLIP models are generally more robust than ImageNet classifiers on 6 visual factors. However, they can be less robust on factors like object pose; In addition, training distribution plays an important role in CLIP robustness against visual factors (Section 4.1).
- CLIP models are biased towards shape when making predictions. However, we have also found that this bias diminishes after fine-tuning on ImageNet and becomes similar to other ImageNet models that are pre-trained on more data. (Section 4.2).
- When trained on the same source, classification accuracy of CLIP models correlates with their OOD detection performance (Section 5).
- CLIP models are not always more calibrated than other ImageNet models, which contradicts existing findings [14]. Our research highlights the impact of training data distribution and quantity on these observations (Section 6).
- Compared to other groups of models, CLIP maintains reasonably good uncertainty estimates under distribution shifts after ID calibration with temperature scaling. (Section 6).

## 2 Related Work

**Robustness** focuses on investigating the resilience of machine learning models to various forms of distribution shift at test time. To this end, a commonly used approach is to introduce artificial transformations onto images, such as, style transfer [15], corruptions and perturbations [16, 17]. Moreover, many real-world datasets are introduced to assess model robustness under a natural distributional shift [3–7]. For instance, Idrissi et al. [13] propose ImageNet-X by relabelling ImageNet validation set to provide detailed labels for naturally occurring factors such as different pose, background and lighting, to identify models' underlying failure patterns.

**OOD detection** targets at identifying test data that do not belong to any of classes modeled in training distribution [18–20]. A large number of methods are proposed for deep learning models, including generative model-based methods [21–28] and discriminative model-based methods [29, 18, 30, 31, 20, 32]. For example, maximum softmax probability [18] is used as the metric to detect OOD samples. Moreover, the above approaches mainly study OOD detection for a task-specific model using only visual information. In contrast, as CLIP models enjoy popularity, zero-shot OOD detection [12], is proposed, where the objective becomes filtering out input from the task of disinterest.

**Predictive uncertainty** aims to classify images with calibrated prediction probabilities so as to match the empirical frequency of correctness [33, 34]. Several works improve uncertainty estimations through post-hoc calibration on validation sets [33, 34]. Moreover, some other works show calibration can be improved by directly applying methods, such as ensembling [35] and pre-training [36]. Ovadia et al. [37] point out that calibration methods become less effective under distribution shift. Minderer et al. [14] suggest that CLIP models are well-calibrated given its accuracy. Based on these observations, this work comprehensively studies the quality of predictive uncertainty given by CLIP.

## 3  Experimental Setup

### 3.1  Models of Interest

**Contrastive language-image pre-training models:** We use 51 **zero-shot CLIP models (CLIP)** and 32 **ImageNet fine-tuned CLIP models (CLIP-FT)**. They have different visual encoders, including slightly modified ResNet [38], ConvNeXt [39], and ViT [40]. There are two training sources (LAION [41] and WIT [1]) and multiple sizes of training datasets from 80 million to 2 billion. For the CLIP-FT models, the vision tower of CLIP is fine-tuned on ImageNet-1K. We consider two fine-tuning procedures, one directly fine-tuned on ImageNet-1K [42], and the other first fine-tuned on ImageNet-12K, a subset of ImageNet-22K before fine-tuning on ImageNet-1K. Unless specified, we use the default prompt template by [1] for zero-shot CLIP models.

**Compared models:** we use 127 ImageNet models with various architectures, including Convolutional Neural Networks (*e.g.*, ResNet [38] and ConvNeXt [39]), Vision Transformers (*e.g.*, ViT [40] and Swin [43]) and all-MLP architectures [44, 45] (*e.g.*, MLP-Mixer [45]). Following [46], we divide them into three categories: **(i) Standard Models.** This group consists of models supervised on the ImageNet training set. **(ii) Contrastive learning models.** This category contains 8 models pre-trained by contrastive learning. There are 6 training algorithms investigated, including InsDis [47], MoCo [48], SimCLR [49]; **(iii) Pre-trained on more data.** This group contains models pre-trained on a significantly larger dataset (*e.g.*, ImageNet-21K) than the ImageNet training set. All the above models, including CLIP, are publicly available on TIMM [50] and OpenCLIP [51]

### 3.2  Test Sets and Metrics

**Robustness.** We first pinpoint failure patterns of models by testing on ImageNet-X [13], which is relabelling of ImageNet validation by 16 naturally occurring factors. This work mainly considers 10 factors labelled with a sufficient number of test samples: *Pose*, *Background*, *Pattern*, *Color*, *Smaller*, *Shape*, *Partial View*, *Subcategory*, *Texture* and *Larger*. The metric is accuracy, and high is better. We evaluate on cue-conflict stimuli and Stylized-ImageNet [15] to measure model bias towards shape.

**OOD Detection.** We use large-scale OOD detection benchmark which is build up on ImageNet: in-distribution (ID) ImageNet *v.s.* {iNaturalist [52], SUN [53], PLACES [54], TEXTURE [55] and ImageNet-O [7] } (OOD). The metrics are the area under the receiver operating characteristic curve (AUROC) and the higher is better; false positive rate (FPR@95) when true positive rate is at 95% and a lower score means better performance.

**Calibration.** We study ID and OOD datasets, where ImageNet validation is ID dataset and OOD datasets are: ImageNet-V2 [3], ImageNet-Rendition [5], ImageNet-Adversarial [7], ImageNet-Sketch [4], ObjectNet [6] and ImageNet-Vid-Robust [56]. Metrics are estimated calibration error (ECE) [57] and negative log likelihood (NLL). A lower ECE or NLL indicates better calibration.

### 3.3  Analytical Methodology

In our endeavor to understand the underlying factors that influence the performance of CLIP models, we delve into six primary aspects: 1) training distribution, evaluating the effect of data source; 2) model architecture, looking into the potential effects of different structural choices on model performance; 3) dataset quantity, probing the interplay between the amount of data available for training and the model's efficiency; 4) contrastive loss, understanding its specific role in training dynamics 5) fine-tuning, and 6) test-time prompt, assessing the impact of prompts during the evaluation on model outputs. We follow the analytical methodology of seminal work [46] and a series of following works like [8, 11, 58]) to study the influential factor. Specifically, within the performance trends observed across all models, any factor causing a deviation from these trends is recognized as influential. Notably, in our research, we mainly emphasize and discuss such influential factors within each facet of our investigation.

## 4  Visual Factor-Level Robustness

The unprecedented robustness of CLIP models has spurred intense research efforts to identify the underlying factors responsible for their performance under distribution shifts. Recent studies provide

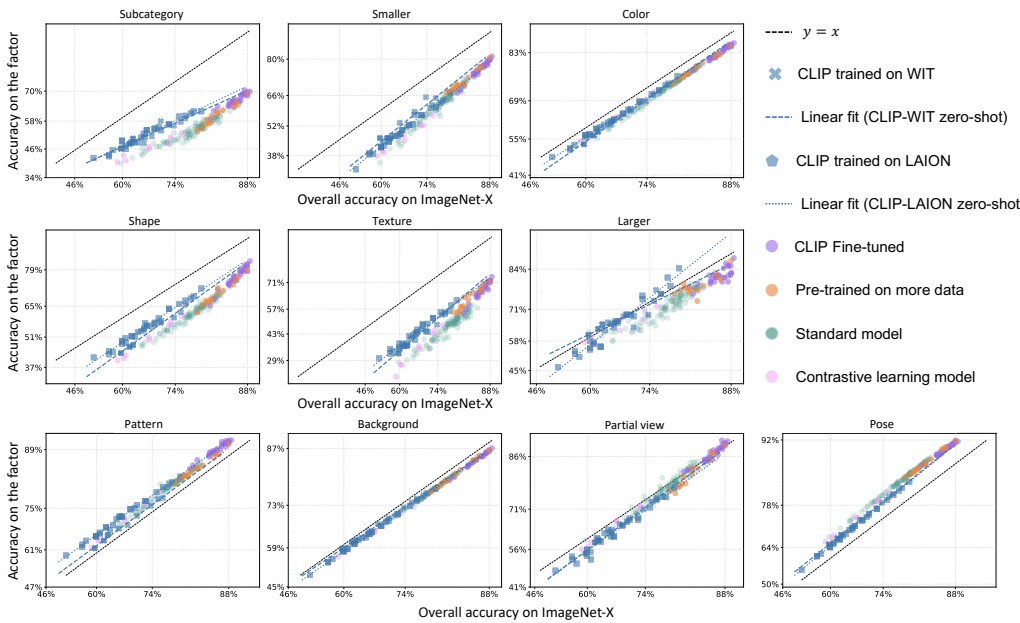

Figure 1: **The models performance on the subset of ImageNet-X annotated with a given visual factor (y-axis) to their overall accuracy on the whole ImageNet-X (x-axis).** Each point represents a model. The x-axis and y-axis are probit transformed following [46]. The black dashed line represents the ideal robust models whose performance on each visual factor is the same as the overall performance. The blue straight lines are fit with robust linear regression [59]. We include models supervised on ImageNet-1K, pre-trained on more data, contrastive learning models, CLIP models trained on two data distributions and their fine-tuned counterparts. We find that CLIP are generally more robust on six out of ten factors, but are less robust against *Pose* than other groups of models.

valuable insights on the design of training source [11, 8]. Our research builds upon previous findings on the robustness of CLIP models and focuses on the potential failure types of the model. Instead of solely measuring overall accuracy across distributions, we investigate the behavior of CLIP models when faced with varying visual factors such as *Pose* and *Background*.

## 4.1 CLIP Models Generally Exhibit Better Factor-Level Robustness Than Other Models

**Factor-level effective robustness**. In our study, we extend the concept of overall effective robustness [46] to visual factor-level. Specifically, it measures a model's ability to achieve higher accuracy on the subset annotated by a specific visual factor compared to what is expected based on its overall accuracy on ImageNet-X. Figure 1 displays the accuracy on the subset annotated by a specific visual factor relative to the overall accuracy on ImageNet-X.

**CLIP models are generally more robust than other ImageNet models on 6 out of 10 visual factors.** Figure 1 highlights several insights into the factor-level robustness of CLIP models. First, we find that CLIP models are more robust than other models on six out of ten visual factors, including *Subcategory*, *Smaller*, *Color*, *Shape*, *Texture*, and *Larger*. Specifically, CLIP models exhibit higher factor-level effective robustness than other models on each of these factors. Second, we observe that CLIP models are less robust than other models on *Pose* and *Partial View* factors. Third, CLIP models show a similar trend to other models on the *Background* factor. In addition, Idrissi et al. [13] observe that data augmentations can improve robustness to related factors, but with spill-over effects to unrelated factors. We speculate that the data augmentations used for training CLIP models may introduce the similar effects.

**Training distributions lead to different trends in CLIP models.** The choice of training distribution impacts factor-level robustness of CLIP models. Specifically, we find that training on different datasets (*i.e.*, LAION and WIT) forms distinct trends on each visual factor for CLIP, and there is no single training source that always leads to higher factor-level robustness than another. For instance,

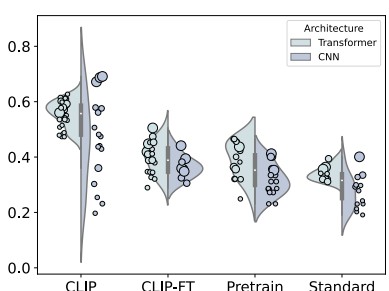

Figure 2: **Shape bias analysis** of CLIP, CLIP fine-tuned (CLIP-FT), models pre-trained on more data (Pretrain), and standard models. Large points mean larger models within the group. We observe that CLIP models are more shape-biased.

| Source | Backbone | Shape bias | IN-Val | SIN |
|---|---|---|---|---|
| LAION | ViT/H-14 (336/224) | 0.42 /**0.51** | **0.89** /0.88 | 0.28 /**0.32** |
| | ViT/L-14 (336/224) | 0.41 /**0.47** | **0.88** /**0.88** | 0.27 /**0.31** |
| | ViT/B-16 (384/224) | 0.35 /**0.43** | **0.87** /0.86 | 0.23 /**0.25** |
| | ViT/B-32 (384/224) | 0.33 /**0.45** | **0.85** /0.83 | 0.21 /**0.22** |
| | ConvNeXt-B (384/224) | 0.31 /**0.38** | **0.87** /0.86 | 0.17 /**0.21** |
| WIT | ViT/L-14 (336/224) | 0.39 /**0.45** | **0.88** /**0.88** | 0.24 /**0.30** |
| | ViT/B-16 (384/224) | 0.35 /**0.41** | **0.87** /0.86 | 0.22 /**0.23** |

Table 1: **The influence of input resolution when fine-tuning CLIP on Shape bias and ImageNet-Val(idation) and Stylized ImageNet (SIN) accuracy.** The higher value in a model pair is in bold. With same backbone, the model fine-tuned with a larger input resolution is more accurate on IN-Val but less shape-biased and less accurate on SIN.

we observe that CLIP models trained on LAION demonstrate higher robustness on *Shape* factor than those trained on WIT, while this reverses for *Background* and *Pose* factors. The results show a mixed observation on *Large* factor. Furthermore, we further point out that CLIP models trained on different subsets of LAION *i.e.*, LAINON-80M, LAION-400M, and LAION-2B) follow the same trend. The above observations highlight the importance of the choice of training source in determining not only the overall accuracy but also the factor-level behaviors of CLIP models. This suggests that visual factor-level robustness should be considered when designing the training source for CLIP models.

**CLIP fine-tuned models perform slightly better than more data pre-trained models.** We compared CLIP fine-tuned models (CLIP-FT) with other models that are pre-trained on more data and find that CLIP-FT shows improvement in overall accuracy and robustness on visual factors of *Subcategory*, *Shape*, and *Pattern*. However, no additional robustness gain is observed on other factors. Additionally, CLIP-FT models outperform zero-shot CLIP on variations such as *Pattern* and *Partial View*, indicating their superiority in handling visual factors. It would be intriguing to explore fine-tuning techniques that maintain or improve the factor-level robustness of zero-shot CLIP.

### 4.2 Texture Bias *v.s.* Shape Bias

**CLIP exhibits a shape bias.** We conduct experiments using the cue-conflict stimuli dataset [15] to investigate the presence of shape bias in the model's predictions. Shape bias, in this context, refers to the proportion of accurate predictions made based on object shapes. Figure 2 presents a visualization of the shape bias exhibited by the models, which are grouped according to their training methods (zero-shot, CLIP finetuning, more data pre-trained, and standard training) and architecture (transformer versus CNN). Our findings indicate that, among the four training methods, CLIP models are more likely to make predictions based on shape compared to the other three groups. Furthermore, while the transformer is reported to have a stronger shape bias than CNN [60, 61], we observe that CLIP using CNN as the vision encoder also exhibit a strong shape bias.

**Model size solely does not explain the shape bias of CLIP**. We further observe that larger CLIP models do not necessarily have higher shape bias than smaller-size ones. For example, both trained on LAION-80M, CLIP-ViT/L-14 has $0.54$ shape bias, which is $0.09$ lower than CLIP-ViT/B-32. This implies that the shape bias of CLIP models cannot be attributed solely to model size. Based on the above observations, we speculate that the shape bias of CLIP may be attributed to its objective, which involves training the model to associate text and image pairs.

**CLIP models have a tendency towards texture bias after fine-tuning**. Our study reveals that shape bias in CLIP weakens after fine-tuning on ImageNet. Moreover, the fine-tuned CLIP models exhibit a shape bias comparable to models that are pre-trained on larger datasets. This finding is consistent when using transformer and CNN as visual encoder. Moreover, these results illustrate that fine-tuning discards the shape-biased property of zero-shot CLIP, which may affect model robustness [62, 15].

**Larger input image resolution during fine-tuning of CLIP results in a stronger bias towards texture.** In Table 1, we observe that an input resolution during fine-tuning is a important factor to shape bias: increasing the input resolution during fine-tuning leads to better performance on ImageNet

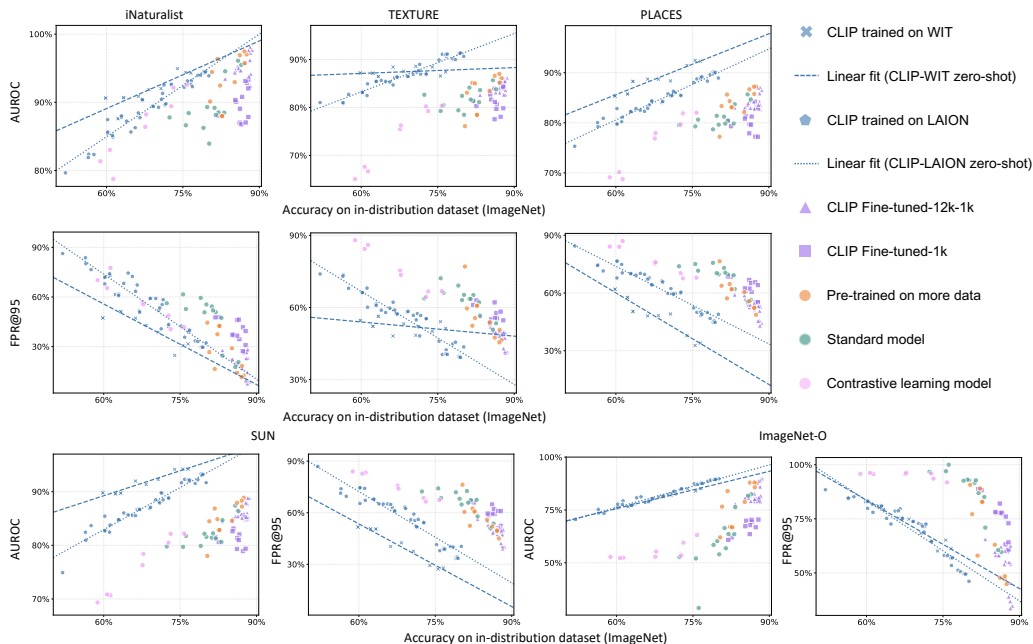

Figure 3: **OOD sample identification capability of models *vs*. ID dataset classification accuracy.** The OOD detection ability is measured by AUROC (↑) and FPR@95 (↓). Each point represents a model. We plot the results on iNaturalist, SUN, PLACES, TEXTURE and ImageNet-O. The blue straight lines are fit with robust linear regression [59]. We observe that training distribution has a greater impact than training dataset quantity on the OOD detection performance of CLIP. Moreover, after additionally fine-tuning on ImageNet-12K, CLIP are generally better at detecting OOD samples than those directly fine-tuned on ImageNet-1K.

validation but also results in more texture-biased models with lower accuracy on Stylized-ImageNet. Across seven pairs of experiments and two training sources, we observe this pattern consistently. Given that input resolution is a crucial model dimension [63–65], it would be interesting to study its effects on shape bias beyond classification accuracy when devising scaling strategies.

## 5 Out-of-Distribution Detection

Zero-shot CLIP allows for a flexible definition of in-distribution (ID) classes without re-training the model. Namely, they can conduct zero-shot OOD detection. The current findings suggest that zero-shot CLIP models are competitive with other state-of-the-art models [12, 66]. Based on this finding, we conduct an extensive analysis to determine whether the purported benefits persist across various training sources, subsets, and network architectures. In the experiments, for zero-shot CLIP models, we utilize maximum concept matching [12] to detect OOD data. For models that are trained or fine-tuned on ImageNet-1K, we employ maximum softmax score [18] for OOD detection.

**For CLIP models from the same source, their ID accuracy correlates with OOD detection performance.** Our study includes CLIP models trained on two sources (WIT and LAION). Given the same training source, our study, conducted across five challenging OOD scenarios, reveals a strong correlation between the ID accuracy of zero-shot CLIP models and their OOD detection performance (measured by AUROC and FPR@95). This observation suggests that the zero-shot classification accuracy of CLIP models on ID data can serve as a reliable indicator of their OOD detection performance. In contrast, such a trend is not as strong for standard models and more data-pre-trained models. Additionally, CLIP-FT models fine-tuned on ImageNet-1K do not exhibit such a clear correlation.

**Training source influences the trend of CLIP.** Upon closer examination of the training distribution, we have observed that the correlation trend between ID accuracy and OOD detection performance is

largely dependent on the training source. As illustrated in Figure 3, our research shows two distinct trends between CLIP models trained on WIT and those trained on LAION. Moreover, with the same ID accuracy, CLIP models trained on WIT exhibit superior OOD detection performance compared to their counterparts trained on LAION on three OOD scenarios. This further indicates the importance of training source selection for CLIP. When developing dataset curation methods, it is valuable to investigate the influence of training sources on OOD detection performance.

**Fine-tuning procedure influences the OOD detection ability of CLIP.** First, we point out that fine-tuning enhances classification performance of CLIP, but this improvement does not necessarily translate to better OOD detection accuracy. Some CLIP-FT models even achieve worse OOD detection performance than Zero-shot CLIP models. Our analysis of CLIP-FT reveals a distinction between two groups of CLIP-FT, based on their fine-tuning procedures: the first group is fine-tuned solely on ImageNet-1K, while the second group undergoes additional fine-tuning on ImageNet-12K. We observe that this additional fine-tuning procedure has a substantial impact on the model's ability to detect OOD examples. As depicted in Figure 3, despite not leading to an improvement in classification accuracy, CLIP-FT models with additional fine-tuning on ImageNet-12K show better OOD detection performance across all OOD scenarios. As future work, it is valuable to investigate this observation further and explore alternative fine-tuning procedures that yield improved OOD detection performance. Moreover, exploring impacts of fine-tuning datasets other than ImageNet-1K and ImageNet-12K would be another interesting direction.

# 6    Prediction Uncertainty

In order to better understand the well-calibrated phenomenon of zero-shot CLIP models reported by Minderer et al. [14], our research systematically analyzes the calibration behavior of CLIP models under various training conditions. Specifically, we examine the calibration performance of CLIP models trained on different training distributions, varied training set sizes, and different architectures. Furthermore, we also investigate the calibration performance of CLIP models after fine-tuning to gain a better understanding of their overall performance.

## 6.1    Zero-Shot CLIP Models Are Not Consistently More Calibrated Than Other Models

**Both training data distribution and quantity impact CLIP's calibration.** Figure 4 presents the model calibration of CLIP models in relation to classification accuracy under distribution shifts. We observe that CLIP models trained on different distributions or quantities are not consistently grouped together. For example, CLIP models trained on WIT and LAION tend to cluster in separate regions. Moreover, when training CLIP models on different subsets of the LAION dataset, models with similar classification accuracy can exhibit varying levels of calibration performance. It would be interesting to further check the impacts of data curation techniques on CLIP calibration performance.

While CLIP models are generally reported to have superior calibration compared to other models [14], our observations reveal that this finding does not always hold. Particularly, we notice that CLIP models trained on LAION-80M dataset exhibit much lower calibration performance when compared to standard models. The observation of [14] is primarily made on CLIP models trained on WIT. However, when we broaden our perspective to include the alternative training distribution provided by LAION and its various subsets, our observations become varied. This emphasizes the significance of careful training source design for CLIP. Furthermore, it suggests that when evaluating dataset curation, it is crucial to consider its impact on the calibration performance of CLIP models.

**CLIP fine-tuned models exhibit a trade-off between calibration and classification.** On each test set in Figure 4, we consistently observe that after fine-tuning, CLIP models tend to have higher classification accuracy and lower calibration error. It is worth noting that additionally fine-tuning on ImageNet-12K does not alter this phenomenon, in contrast to its impact on OOD detection. Moreover, other model groups, including those pre-trained on more data, do not exhibit a trade-off between calibration and classification. We also observe some fine-tuned CLIP models achieve better calibration compared to their zero-shot counterparts before calibration.

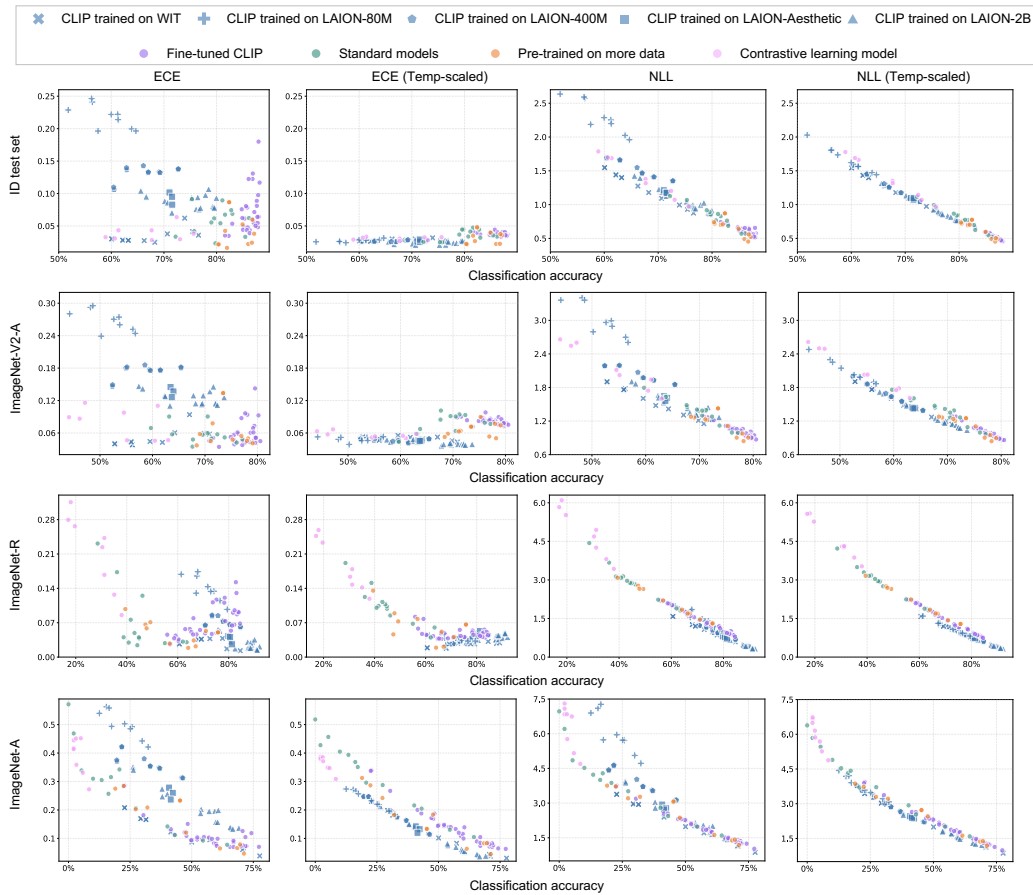

Figure 4: **Model calibration performance with respect to their classification accuracy**. We report results on in-distribution test set, ImageNet-V2-A, ImageNet-R and ImageNet-A. Two metrics are considered: ECE (↓) and NLL (↓), we also include calibration performance after calibration with temperature scaling. Each point represents a model. We use colors to represent model groups. For zero-shot CLIP, we additionally use shapes to indicate training distribution and quantity. We observe that CLIP models could be less calibrated than standard models. The training distribution and quantity are the key factors influencing the calibration performance of CLIP models. Temperature scaling reveals a consistent trend of CLIP models, and they tend to lie on a distinct trend from other models.

## 6.2 Temperature Scaling Reveals Well-Calibrated Properties of Zero-Shot CLIP Models

Post-hoc calibration can be adopted to remedy over- or under-confidence. Here, we use temperature scaling [33] to calibrate model predictions. Following the protocol in [67], we divide the validation set of ImageNet into two halves: one for temperature scaling (ID calibration set), and the other one for ID test. We report results on both ID and OOD test sets.

**Calibration performance of CLIP models after temperature scaling (Temp-scaled) correlates with their classification accuracy.** In Figure 4, we explore how temperature scaling affects different groups of CLIP models. These groups are categorized based on the amount and source of their training data. After applying temperature scaling and evaluating using the NLL metric, we observe a consistent pattern among these CLIP groups. What is intriguing is that, after temperature scaling, for models with similar image classification accuracy, zero-shot CLIP models achieve better calibration performance compared to other models, including their fine-tuned counterparts. This trend persists across various testing scenarios, encompassing ID, OOD, and when assessing calibration using both NLL and ECE metrics.

**ID calibration of CLIP models transfers to OOD test sets**. While it has been reported by Ovadia et al. [37] that ID calibration often struggles to generalize under distribution shifts, our study reveals

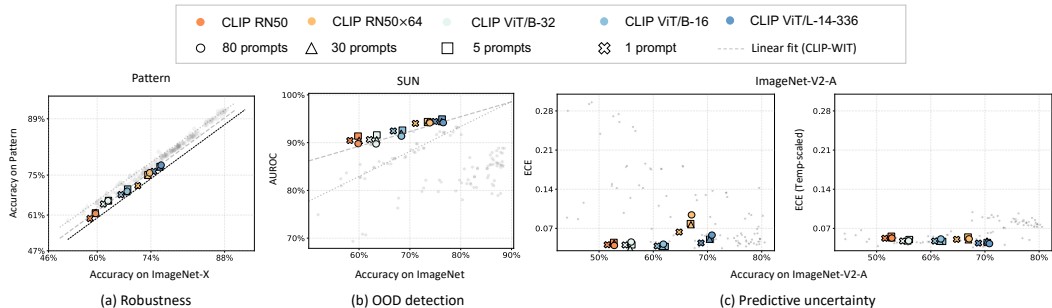

Figure 5: **Influence of test time prompt on CLIP on robustness to visual factors, OOD detection, and predictive uncertainty**. We include five CLIP models trained on WIT. We use different colors to denote different model architectures and utilize various shapes to represent deployed prompt sets. The dashed grey line is fit with robust linear regression [59] by the original CLIP-WIT models using 80 prompts. We see that the prompts of sizes 1, 5 and 30 decrease the classification performance of CLIP, but may not change the visual factor robustness of CLIP.

a promising phenomenon for CLIP models. Specifically, after calibrating CLIP models on ID calibration set, we observe improved calibration results on OOD test sets. For example, on ImageNet-A, the calibration errors of CLIP models decrease after temperature scaling, unlike other models that do not clearly exhibit such improvement. This indicates that CLIP models are relatively easier to calibrate across diverse distributions, highlighting their potential for robust and reliable applications.

## 7 Discussion on Influence of Test Time Prompts

So far in the experiments, we utilize the prompt provided by [1]. In this section, we investigate the effect of test time prompts on three safety objectives. We additionally examine another three sets of prompts, one of size 1: "`a photo of a {label}`", a set of 5 prompts used by [12], and a set of 30 prompts. We conduct the experiment on five CLIP models: RN50, RN50×64, ViT-B/16, ViT-B/32 and ViT/L-14-336px trained on WIT. Figure 5 shows the performance of CLIP models using different sets of prompts on three safety objectives.

We show that utilization of few prompts (*e.g.*, one prompt) generally leads to a decrease in overall classification performance. However, the impact on the three properties is mixed. First, when evaluating factor-level robustness on Pattern, we find the adoption of different prompts does not alter the robustness: models still adhere to the linear trend established by CLIP trained on WIT using 80 prompts. Second, for OOD detection, using 5 prompts yield higher OOD detection performance than with 80 prompts on SUN. Also, for calibration, using fewer prompts (*e.g.*, only one prompt) achieves lower calibration error than using all 80 prompts. This raises an important question: how to tune prompts to achieve better classification accuracy as well as better calibration and OOD detection performance? It would be interesting to study this when conducting prompt learning [68–70].

## 8 Conclusion and Discussion

Our research provides a valuable contribution to the ongoing discourse surrounding the effectiveness of CLIP models in the context of robustness to visual factors, OOD detection, and the reliability of uncertainty estimation. In pursuit of these objectives, we conduct extensive experiments encompassing three critical tasks and perform comparative analyses between CLIP models and various groups of models. Our observations provide valuable insights into the advantages of CLIP models. First, CLIP models demonstrate superior visual factor-level robustness compared to other ImageNet models. Furthermore, while maintaining comparable accuracy on in-distribution dataset, CLIP models also exhibit competitive performance in OOD detection across commonly used benchmarks such as iNaturalist and ImageNet-O. Lastly, CLIP models are relatively easier to calibrate across diverse distributions. Furthermore, our study highlights the significance of training source design, as it profoundly influences the behavior of CLIP models across all three objectives.

This work leaves open many interesting directions for future research and we discuss a few. **First**, this work primarily studies CLIP and its fine-tuned models due to its simplicity and effectiveness. We

reckon this work as an anchor point and hope the framework of this analysis could generalize to other image-text foundation models, such as ALIGN [2] and BASIC [71]. **Second**, our study includes two academic training sources, namely WIT and LAION, for CLIP. It is valuable to investigate whether our observations generalize to other training sources. We believe that our study can shed light to and build up the understanding towards the design of multi-modal datasets. **Lastly**, in addition to three safety-critical tasks, there are other important fields to analyze such as mis-classification samples detection [18]. By providing a more detailed and nuanced understanding of the performance of CLIP models, we hope our observation and insights can inform future developments in the field and help to drive progress towards more robust and effective vision-language models.

**Broader impacts.** While much research endeavors that aim to enhance the performance of machine learning models could be leveraged for negative societal applications, we believe that this paper points towards a positive direction for the development of machine learning methods in broader society. Particularly, our study understands the influence of training set on CLIP performance on robustness, out-of-distribution detection, and predictive uncertainty. A better understanding is beneficial in establishing trustworthy machine learning systems and high-quality multi-modal datasets for training.

**Acknowledgement.** We thank all anonymous reviewers and ACs for their constructive comments and valuable suggestions in improving this paper.

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
