# Supplementary for
# A Closer Look at the Robustness of Contrastive Language-Image Pre-Training (CLIP)

In Section A, we introduce the experimental details including the training details, datasets and computation resources. Then, we report the full results of factor-level robustness in Section B; In Section C, we compare the OOD detection performance of zero-shot CLIP using maximum concept matching [1] and other models using maximum logit [2]; Lastly, we show the calibration performance of models on ImageNet-Rendition [3], ObjectNet [4] and ImageNet-Vid-Robust [5] (Section D).

## A  Experimental Setup

### A.1  Datasets

We carefully check the licenses of all datasets and list the open sources to them.

**Datasets used in factor-level robustness:**

ImageNet [6] (https://www.image-net.org);
ImageNet-X [7] (https://github.com/facebookresearch/imagenetx);

| Factor | Number of samples |
|---|---|
| Pose | 16080 |
| Background | 15441 |
| Pattern | 6571 |
| Color | 6476 |
| Smaller | 1473 |
| Shape | 696 |
| Partial View | 678 |
| Subcategory | 614 |
| Texture | 286 |
| Larger | 157 |
| Darker | 125 |
| Object Blocking | 78 |
| Person Blocking | 61 |
| Style | 45 |
| Brighter | 45 |
| Multiple Objects | 40 |
| Overall | 48868 |

Table A-1: **Statistics of ImageNet-X of each factor.**

**Datasets used in out-of-distribution detection:**

iNaturalist [8] (https://www.kaggle.com/c/inaturalist-2021);
SUN [9] (https://groups.csail.mit.edu/vision/SUN/hierarchy.html);

`TEXTURE` [10] (https://www.robots.ox.ac.uk/ vgg/data/dtd/);
`PLACES` [11] (https://github.com/CSAILVision/places365);
`ImageNet-O` [9] (https://github.com/hendrycks/natural-adv-examples);

**Datasets used in predictive uncertainty:**

`ImageNet-V2` [5] (https://github.com/modestyachts/ImageNetV2);
`ImageNet-Adversarial` [9] (https://github.com/hendrycks/natural-adv-examples);
`ImageNet-Rendition` [3] (https://github.com/hendrycks/robustness);
`ImageNet-Sketch` [12] (https://github.com/HaohanWang/ImageNet-Sketch);
`ObjectNet` [4] (https://objectnet.dev/download.html);
`ImageNet-Vid-Robust` [5] (https://github.com/modestyachts/imagenet-vid-robust-example).

## A.2 Models Included in Experiments

### A.2.1 Zero-shot CLIP

We use the zero-shot CLIP models provided in OpenCLIP [13]. They are listed as follows in the pattern (`architecture`, `source`):

(RN50, openai)

(RN50-quickgelu, openai)

(RN101, openai)

(RN101-quickgelu, openai)

(RN50×4, openai)

(RN50×16, openai)

(RN50×64, openai)

(ViT-B-32, openai)

(ViT-B-32-quickgelu, openai)

(ViT-B-16, openai)

(ViT-L-14, openai)

(ViT-L-14-336, openai)

(ViT-B-32, laion400m_e31)

(ViT-B-32, laion400m_e32)

(ViT-B-32, laion2b_e16)

(ViT-B-32, laion2b_s34b_b79k)

(ViT-B-32-quickgelu, laion400m_e31)

(ViT-B-32-quickgelu, laion400m_e32)

(ViT-B-16, laion400m_e31)

(ViT-B-16, laion400m_e32)

(ViT-B-16, laion2b_s34b_b88k)

(ViT-B-16-plus-240, laion400m_e31)

(ViT-B-16-plus-240, laion400m_e32)

(ViT-L-14, laion400m_e31)

(ViT-L-14, laion400m_e32)

(ViT-L-14, laion2b_s32b_b82k)

(ViT-H-14, laion2b_s32b_b79k)

(ViT-g-14, `laion2b_s12b_b42k`)

(ViT-g-14, `laion2b_s34b_b88k`)

(ViT-bigG-14, `laion2b_s39b_b160k`)

(`convnext_base`, `laion400m_s13b_b51k`)

(`convnext_base_w`, `laion2b_s13b_b82k`)

(`convnext_base_w`, `laion2b_s13b_b82_augreg`)

(`convnext_base_w`, `laion_aesthetic_s13b_b82k`)

(`convnext_base_w_320`, `laion_aesthetic_s13b_b82k`)

(`convnext_base_w_320`, `laion_aesthetic_s13b_b82k_augreg`)

(`convnext_large_d`, `laion2b_s26b_b102k_augreg`)

(`convnext_large_d_320`, `laion2b_s29b_b131k_ft`)

(`convnext_large_d_320`, `laion2b_s29b_b131k_ft_soup`)

(`convnext_xxlarge`, `laion2b_s34b_b82k_augreg`)

(`convnext_xxlarge`, `laion2b_s34b_b82k_augreg_rewind`)

(`convnext_xxlarge`, `laion2b_s34b_b82k_augreg_soup`)

Nine models provided by Cherti et al. [14]: https://github.com/LAION-AI/scaling-laws-openclip:

(ViT-B-32, `Model-B-32_Data-80M_Samples-3B_lr-5e-4_bs-32k.pt`)

(ViT-B-32, `Model-B-32_Data-80M_Samples-13B_lr-5e-4_bs-32k.pt`)

(ViT-B-32, `Model-B-32_Data-80M_Samples-34B_lr-1e-3_bs-88k.pt`)

(ViT-B-16, `Model-B-16_Data-80M_Samples-3B_lr-1e-3_bs-88k.pt`)

(ViT-B-16, `Model-B-16_Data-80M_Samples-13B_lr-1e-3_bs-88k.pt`)

(ViT-B-16, `Model-B-16_Data-80M_Samples-34B_lr-1e-3_bs-88k.pt`)

(ViT-L-14, `Model-L-14_Data-80M_Samples-3B_lr-1e-3_bs-88k.pt`)

(ViT-L-14, `Model-L-14_Data-80M_Samples-13B_lr-1e-3_bs-88k.pt`)

(ViT-L-14, `Model-L-14_Data-80M_Samples-34B_lr-1e-3_bs-88k.pt`)

### A.2.2 Fine-tuned CLIP on ImageNet

We consider the fine-tuned CLIP models provided in TIMM [15]. They are listed as follows:

`vit_huge_patch14_clip_336.laion2b_ft_in12k_in1k`

`vit_huge_patch14_clip_224.laion2b_ft_in12k_in1k`

`vit_large_patch14_clip_336.laion2b_ft_in12k_in1k`

`vit_large_patch14_clip_224.laion2b_ft_in12k_in1k`

`vit_base_patch16_clip_384.laion2b_ft_in12k_in1k`

`vit_base_patch16_clip_224.laion2b_ft_in12k_in1k`

`vit_base_patch32_clip_448.laion2b_ft_in12k_in1k`

`vit_base_patch32_clip_384.laion2b_ft_in12k_in1k`

`vit_base_patch32_clip_224.laion2b_ft_in12k_in1k`

`vit_huge_patch14_clip_224.laion2b_ft_in1k`

`vit_large_patch14_clip_336.laion2b_ft_in1k`

`vit_large_patch14_clip_224.laion2b_ft_in1k`

```
vit_base_patch16_clip_384.laion2b_ft_in1k
vit_base_patch16_clip_224.laion2b_ft_in1k
vit_base_patch32_clip_224.laion2b_ft_in1k
vit_large_patch14_clip_336.openai_ft_in12k_in1k
vit_large_patch14_clip_224.openai_ft_in12k_in1k
vit_base_patch16_clip_384.openai_ft_in12k_in1k
vit_base_patch16_clip_224.openai_ft_in12k_in1k
vit_base_patch32_clip_384.openai_ft_in12k_in1k
vit_large_patch14_clip_224.openai_ft_in1k
vit_base_patch16_clip_384.openai_ft_in1k
vit_base_patch16_clip_224.openai_ft_in1k
vit_base_patch32_clip_224.openai_ft_in1k
convnext_base.clip_laion2b_augreg_ft_in1k
convnext_base.clip_laiona_augreg_ft_in1k_384
convnext_large_mlp.clip_laion2b_augreg_ft_in1k
convnext_large_mlp.clip_laion2b_augreg_ft_in1k_384
convnext_large_mlp.clip_laion2b_soup_ft_in12k_in1k_384
convnext_large_mlp.clip_laion2b_soup_ft_in12k_in1k_320
convnext_base.clip_laion2b_augreg_ft_in12k_in1k_384
convnext_base.clip_laion2b_augreg_ft_in12k_in1k
```

### A.2.3 Other ImageNet Models

The standard models and those which are pre-trained on more data are publicly available on TIMM as well. We refer to the supplementary of Taori et al. [16] and Deng et al. [17] for more details. For models trained by contrastive learning, they are provided by Geirhos et al. [18] in https://github.com/bethgelab/model-vs-human. They are:

PIRL [19]

InsDis [20]

MoCo [21]

MoCoV2 [22]

InfoMin [23]

simclr_resnet50×1, simclr_resnet50×2, simclr_resnet50×4 [24]

### A.3 Computation Resources

PyTorch version is 1.10.0+cu111 and timm version is 0.8.21dev0. All experiment is run on one 3090 and the CPU AMD EPYC 7343 16-Core Processor.

## B  Full Results on Visual Factor-level Robustness

In the main paper, we report the results on 10 visual factors which contain over 150 test samples. Here, we additionally report the visual factor-level robustness on another 6 factors, and they have less than 150 test samples. The statistics of visual factors are summarized in Table A-1. As shown in Figure A-1, on the new 6 visual factors, we observe that CLIP models tend to have higher factor-level

robustness. Moreover, by looking at CLIP models trained on WIT and LAION, we still observe that the training sources lead to different trends in CLIP models.

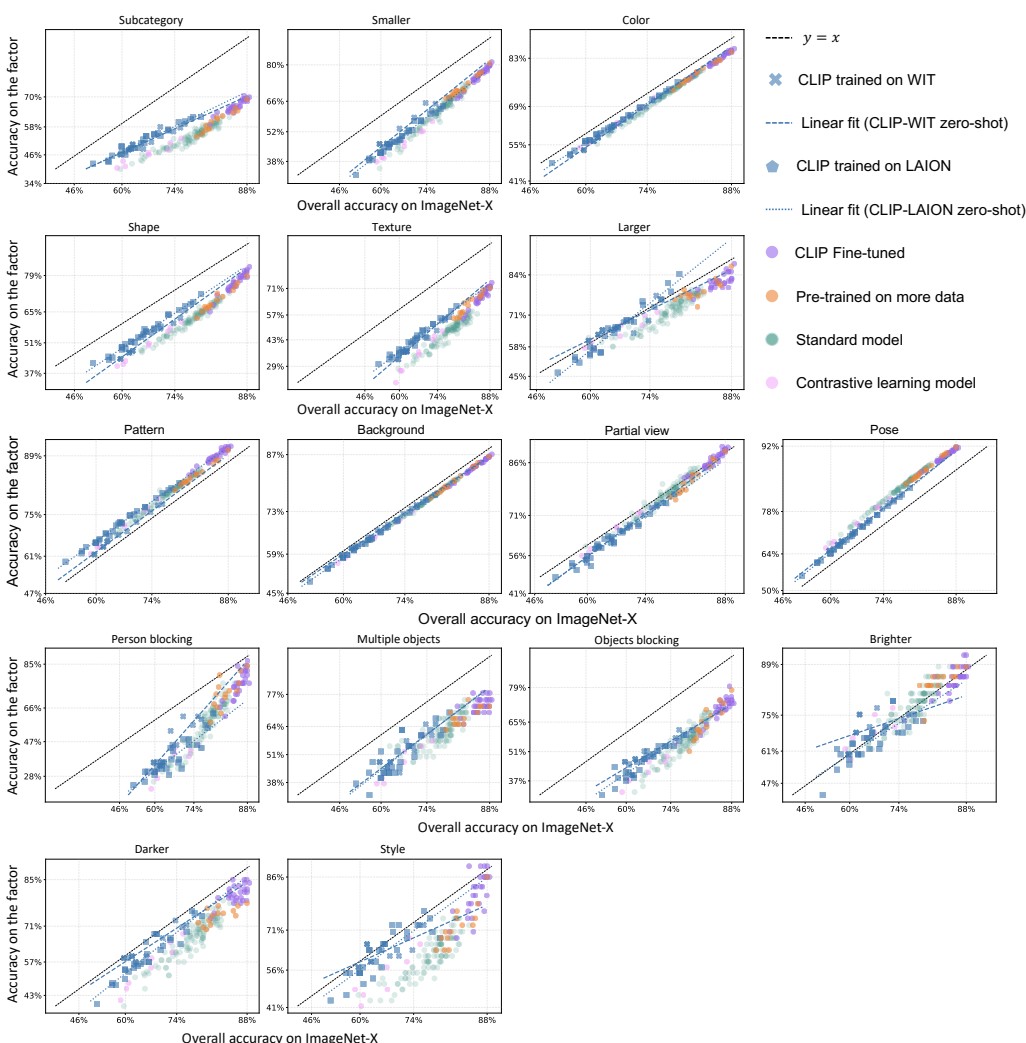

Figure A-1: **Accuracy on a factor *v.s.* overall accuracy on ImageNet-X on 16 factors.** We additionally include another 6 factors: *Person blocking*, *Multiple objects*, *Object blocking*, *Brighter*, *Darker* and *Color*, and the number of images labeled by these factors is no more than 150. We further observe that CLIP models tend to be more robust than other ImageNet models on these factors.

# C  More Results on Out-of-distribution Detection

In the main paper, we use the maximum softmax score as the detection method for the models that are trained or fine-tuned on ImageNet. Here, we instead use maximum logit [2] for them. Zero-shot CLIP models still use maximum concept matching [1].

As shown in Figure A-2, CLIP models remain competitive with other models. Moreover, the observations reported in the main paper (Section 5) still hold. Specifically, 1) CLIP-FT models fine-tuned on ImageNet-1K do not exhibit a clear correlation between OOD detection performance and classification accuracy. 2) With similar in-distribution accuracy, CLIP models tend to achieve competitive or higher OOD detection performance compared to other groups of models. 3) We find that additional fine-tuning on ImageNet-12K substantially improves the model's ability to detect OOD examples compared to those directly fine-tuned on ImageNet-1K.

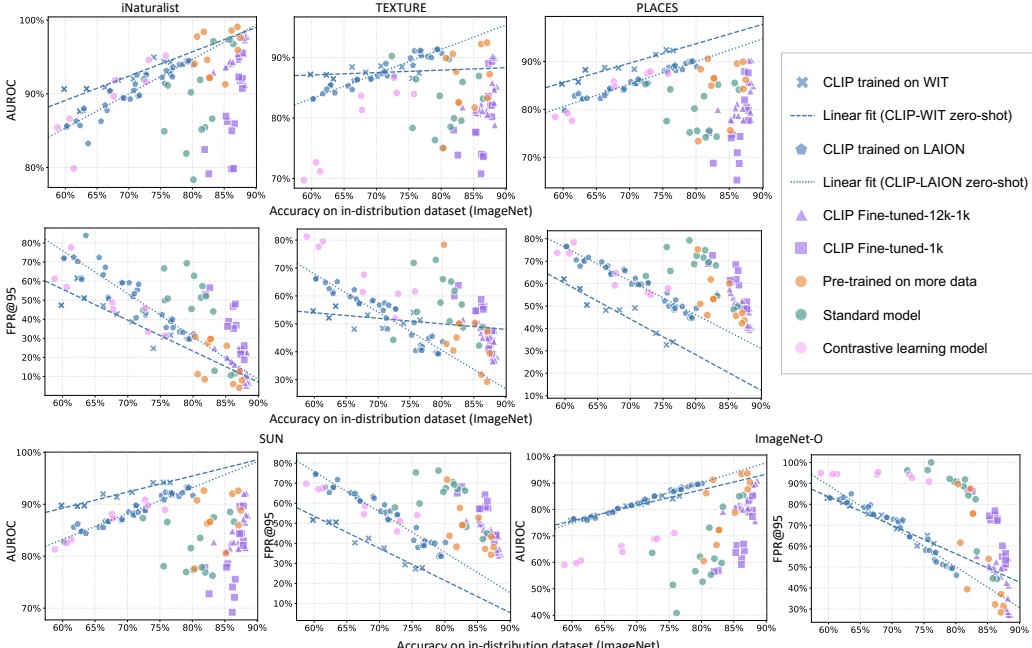

Figure A-2: **Performance of OOD detection *v.s.* accuracy on ImageNet.** The OOD detection ability is measured by AUROC (↑) and FPR@95 (↓). Each point represents a model. We plot the results on iNaturalist, SUN, PLACES, TEXTURE and ImageNet-O. The blue straight lines are fit with robust linear regression [25]. We observe that training distribution has a greater impact than training dataset quantity on the OOD detection performance of CLIP. For CLIP models from the same source, their ID accuracy correlates with OOD detection performance. Moreover, after additionally fine-tuning on ImageNet-12K, CLIP models are generally better at detecting OOD samples than those directly fine-tuned on ImageNet-1K.

Moreover, we include the results of OOD detection performance on NINCO [26] in Figure A-3. We find the observations on this dataset consistent with those on five standard OOD detection benchmarks: 1) for CLIP models from the same source, their ID accuracy correlate with OOD detection; 2) training source influences trends of CLIP; 3) fine-tuning procedure influences OOD detection ability of CLIP. We use following models:

`vit_base_patch16_384, vit_base_patch16_224_21kpre,`

`swinv2-1k, swinv2-22k,`

`deit3-384-1k, deit3_base_patch16_224,`

`xcit_medium_24_p16_224, xcit_medium_24_p16_224_dist,`

```
convnext_base, convnext_base_in22ft1k,

BiT_s, BiT_m,

tf_efficientnetv2_m, tf_efficientnetv2_m_in21ft1k,

resnet50, resnet18.tv_in1k, 'resnet34.tv_in1k',

efficientnet_b0, tf_efficientnet_b1.in1k, tf_efficientnet_b7,

repvgg_b0.rvgg_in1k,

inception_v4.tf_in1k,

mobilenetv2_050.lamb_in1k,

deit3-384-22k, deit3_base_patch16_224_in21ft1k
```

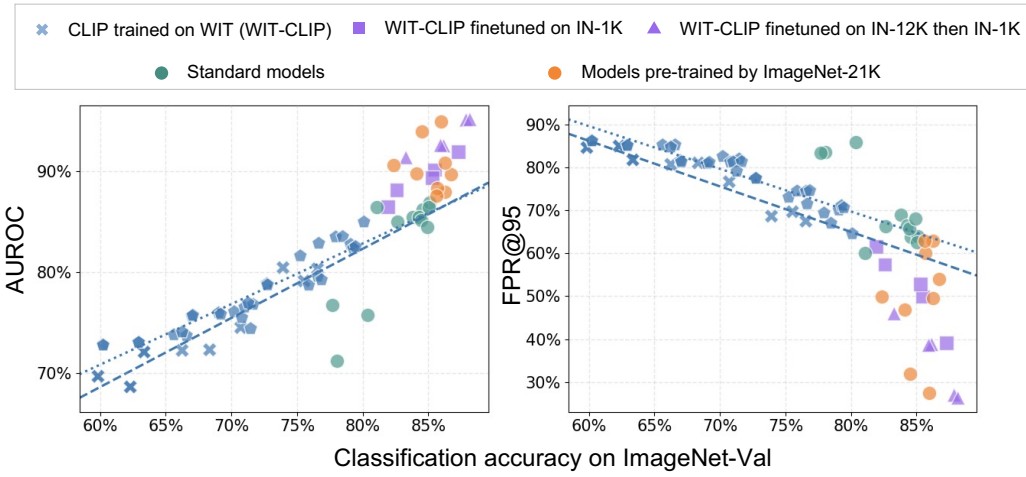

Figure A-3: **New benchmark results on NINCO for OOD detection.** Classification accuracy on ImageNetVal vs. model OOD detection measured by classes mean AUROC (↑) and classes mean FPR@95 (↓). Standard models and models pretrained on IN-21K use MSP.

## D   More Results on Predictive Uncertainty

We show the full results of uncertainty estimation of all models on all datasets: ImageNet-Val (ID test set), ImageNet-V2-A, ImageNet-A, ImageNet-S, ImageNet-R, ObjectNet and ImageNet-Vid-Robust in Figure A-4. We use estimated calibration error (ECE) and negative log-likelihood (NLL) as evaluation metrics. We also report the results after calibration using temperature scaling. As for the calibration, following the protocol in [27], we divide the validation set of ImageNet into two halves: one for temperature scaling (ID calibration set), and the other one for ID test.

## E   Experiments on MAE models

Figure A-5 shows the performance of models pre-trained by MAE[28] on factor level robustness, OOD detection and calibration. MAE is not a contrastive training approach in contrast to MoCo and SimCLRV2. We see that: 1) Visual factor-level robustness: three MAE models lie on the area of models pre-trained on more data; 2) OOD detection: MAE models also lie in the area with other models pre-trained on more data. Some zero-shot CLIP models achieve higher performance than MAE models; 3) Calibration: before TS, MAE models have higher uncertainty estimation performance than CLIP trained on LAION while lower than CLIP trained on WIT. Post TS, CLIP models become better than MAE.

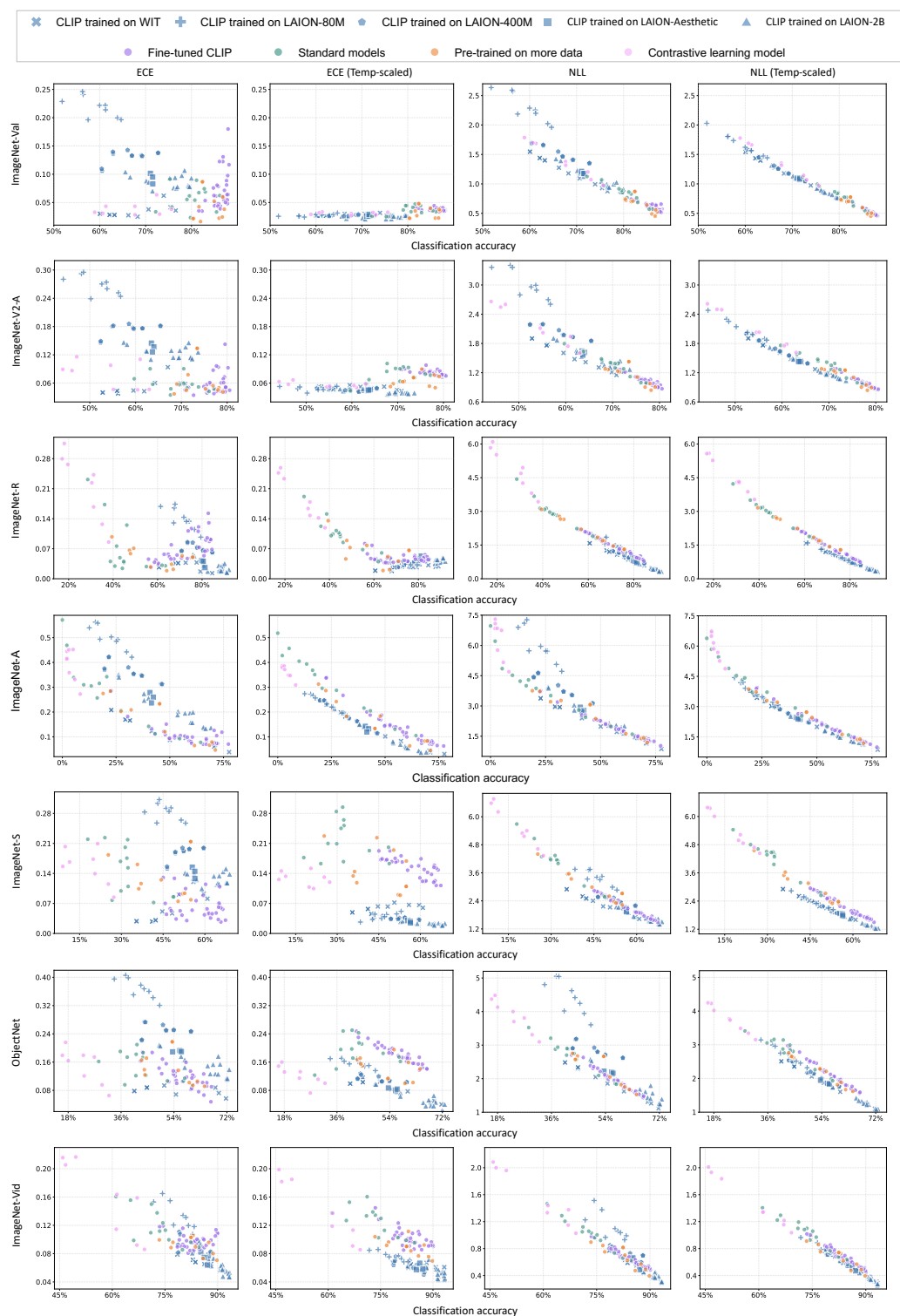

Figure A-4: **Performance of uncertainty estimation *v.s.* classification accuracy.** We report results on 7 datasets, using ECE (↓) and NLL (↓). We also include calibration performance after calibration with temperature scaling. Each point represents a model. We use colors to represent model groups. For zero-shot CLIP models, we use shapes to indicate training distribution and quantity.

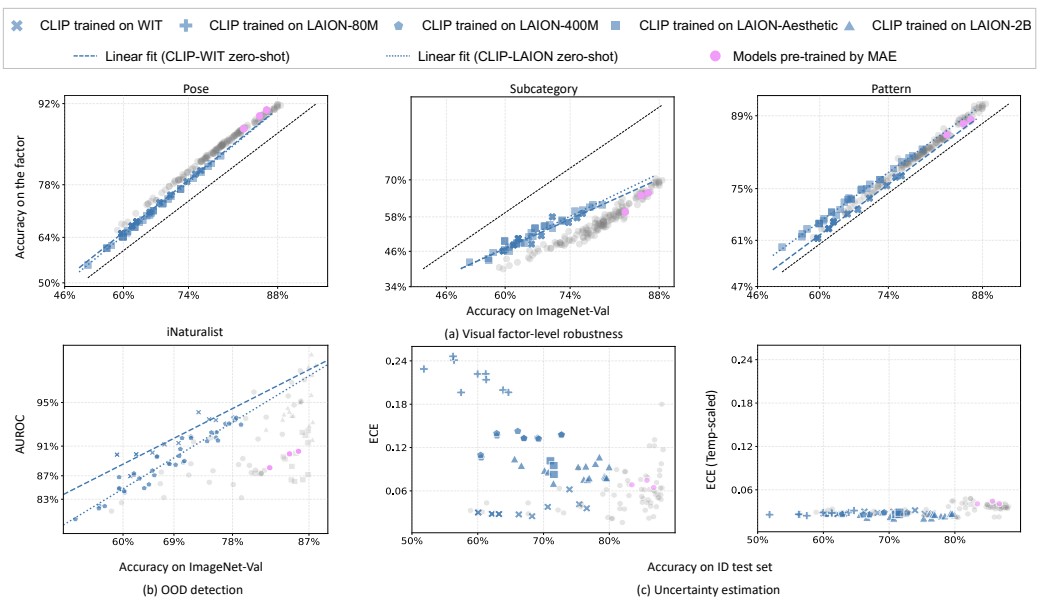

Figure A-5: We evaluate three models pre-trained by MAE. **(a)** The models performance on the subset of ImageNetX annotated with a given visual factor (y-axis) to their overall accuracy on the whole ImageNet-X (x-axis). **(b)** OOD detection capability of models vs. ID dataset classification accuracy. **(c)** Model calibration performance with respect to their classification accuracy.

## F    Analysis on Retrieval performance of Zero-shot CLIPs

Figure A-6 illustrates the image classification accuracy on ImageNet validation set versus their zero-shot retrieval performance of CLIP models on MS-COCO2017. We find that the zero shot retrieval performance correlate with its classification accuracy. CLIPs trained on different dataset distribution lie on different trends. Such correlation shows that the ability of image classification is predictive of their image or text retrieval capability. Interestingly, we observe there are four specific ConvNeXt-based CLIP models that significantly depart from the trend of LAION. We notice that they are trained with a limited random resize crop range (0.9, 1.0), which may hurt the capability of learned embeddings.

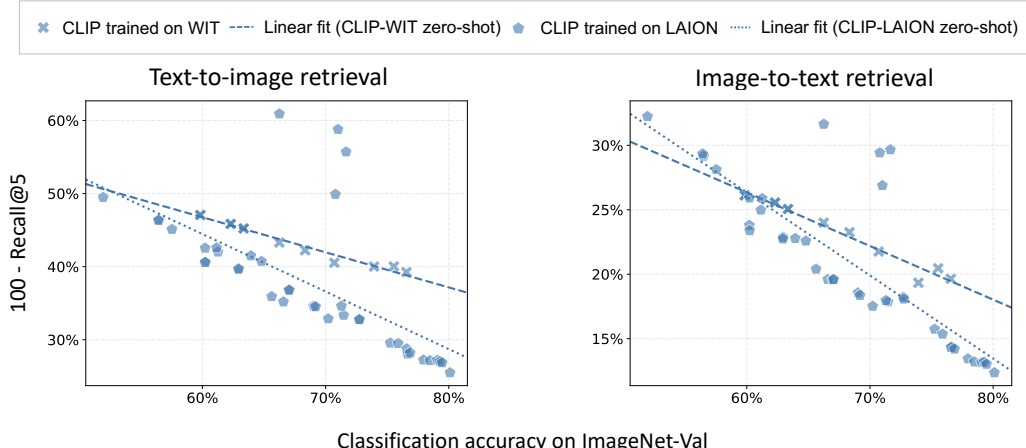

Figure A-6: Classification accuracy vs. retrieval performance measured by (100 – Recall@5) (↓).