# OpenReview forum: "A Closer Look at the Robustness of Contrastive Language-Image Pre-Training (CLIP)"
_NeurIPS.cc/2023/Conference — NeurIPS 2023 poster_

### Official Review · Reviewer_A6cY · 2023-06-16

**Soundness:** 4 excellent
**Presentation:** 3 good
**Contribution:** 3 good
**Rating:** 6
**Confidence:** 4

**Summary:**

This paper performs a comprehensive study of various CLIP models on robustness to different visual factors, out-of-distribution detection, and calibrated uncertainty estimations. A total number of 53 CLIP models trained on different training sources and sizes, and different architectures, with additionally 32 CLIP models fine-tuned on ImageNet are studied.

**Strengths:**

[Originality]

To the best of my knowledge, no previous studies have done such experiments on the 3 aspects to study CLIP models, especially considering various CLIP models trained on different datasets. Some of the observations made in this paper are new and complementary to the previous studies. For example, CLIP models are not robust in all aspects - they are less robust than models trained on ImageNet in a supervised way when poses are changed. Therefore, the findings are novel.

[Significance]

Although most conclusions that can be drawn from the experiments are already known, e.g., CLIP models are more robust than supervised models, some of them are less well-known and may be valuable to the community. For example, CLIP models trained on WIT perform better than those trained on LAION on OOD detection.

[Quality & Clarity]

The quality and presentation of this paper are good. It is easy to understand and follow.



**Weaknesses:**

Some of the discussion on the experiment observations might need more support.

1. Line 177: "The shape bias of CLIP may be attributed to its objective, which involves training the model to associate text and image pairs": in Figure 2, it seems that fine-tuning CLIP models on ImageNet (with supervised objective?) decreases the shape bias. However, it is also possible that the data source (ImageNet) could be the reason. Is it possible to decouple this? Say fine-tune CLIP with the contrastive objective on ImageNet and see if the shape bias stays the same.

2. Line 246: "We notice that CLIP models trained on LAION-80M dataset exhibit lower calibration performance when compared to standard models." Is this fair to compare as CLIP models trained on LAION-80M generally have lower accuracy?

Another minor concern I have is that some results from previous studies are not clearly discussed. For example, in ImageNet-X, it is already observed that "color-jitter augmentation improves robustness to color and brightness, but hurts robustness to pose." Since color-jittering is widely used in the CLIP training, this should be discussed in the paper.

[1] Badr Youbi Idrissi, Diane Bouchacourt, Randall Balestriero, Ivan Evtimov, Caner Hazirbas, Nicolas Ballas, Pascal Vincent, Michal Drozdzal, David Lopez-Paz, and Mark Ibrahim. Imagenet-x: Understanding model mistakes with factor of variation annotations. In International Conference on Learning Representations, 2022.

[Minor]
Line 94: ImageNet 32 fine-tuned CLIP models -> 32 ImageNet fine-tuned CLIP models
Line 141: Small -> Smaller



**Questions:**

I have the following questions that would help me better understand the paper, and I would appreciate the authors' reply to them:

1. Is there any insight on why WIT differs from LAION when serving as the CLIP training set, since I thought both of them were deemed to be a comprehensive snippet of the web image-text pairs?

2. What about the other self-supervised models like MAE when compared with CLIP in terms of robustness? This also helps demystify the factors of datasets and objectives.

3. ImageNet-1k contains 1.3 million images, and ImageNet-21k contains > 20 million images, which are not far away from the smallest LAION dataset considered in the paper. Is it safe to say that when the dataset size is similar, the supervised objective is still better in terms of calibrated uncertainty estimations? How about robustness and OOD detection?

**Limitations:**

The authors discuss the broader impacts in the conclusion section.

---

> ### Author Rebuttal · Authors · 2023-08-09
>
> >Q1: Figure 2, fine-tuning CLIPs on ImageNet (with supervised objective?) decreases shape bias … data source (ImageNet) could be the reason … fine-tune CLIP with contrastive objective to see if shape bias stays the same.
>
> Insightful suggestion. All fine-tuned CLIP models in Fig. 2 use supervised cross-entropy loss. To decouple the effects of contrastive learning and data source, we employed zero-shot CLIP with ViT-B-32 and fine-tuned it on ImageNet by two methods: standard cross-entropy and contrastive loss [a].
>
> The shape bias extents are measured as 0.58, 0.40, and 0.56 for zero-shot CLIP, standard fine-tuned CLIP, and contrastive-loss fine-tuned CLIP, respectively. This indicates ImageNet might not be the primary cause of the shape-bias decrease. Also, we maintain speculation that associating image embeddings with text embeddings could potentially help learn shape-biased models.
> We will include the above discussion in the revised version.
>
> [a] Finetune like you pretrain: Improved finetuning of zero-shot vision models. CVPR’23
>
> >Q2. Is it fair to compare CLIP trained on LAION-80M and standard models as the former ones generally have lower accuracy?
>
> Thanks for raising this discussion. First, it is a common practice to compare uncertainty estimation performance between models with varying accuracy levels, as exemplified by references [b,c,d] and many others.
>
> Second, in the main paper, CLIP models trained on LAION-80M are used to demonstrate that CLIP models do not always exhibit superior calibration compared to other ImageNet models, contrary to existing observations. We further clarify that, at comparable accuracy levels, CLIP models trained on LAION-80M are observed to have higher calibration error (ECE).
>
> Last, we firmly believe in the necessity of a comprehensive evaluation of CLIP models' robustness, driving the primary motivation behind our study. When measuring CLIP models' robustness, we advocate for considering more perspectives alongside the commonly used classification accuracy
>
> We will revise the Line 246-247 to make it clear.
>
> [b] On calibration of modern neural networks, ICML'17
>
> [c] Improving model calibration with accuracy versus uncertainty optimization, NeurIPS'20
>
> [d] Revisiting the Calibration of Modern Neural Networks. NeurIPS'21
>
> >Q3. Some results from previous studies are not clearly discussed … “color-jitter augmentation improves robustness to color and brightness but hurts robustness to pose".
>
> Insightful point. We agree that data augmentation used in CLIP training is crucial for their factor-level robustness. Specifically, data augmentations can improve robustness to related factors, but with spill-over effects to unrelated factors. As the reviewer points out, colour-jittering serves to improve robustness to color and brightness variations while influencing object pose. Similarly, scale-based augmentation could facilitate the learning of scale-invariant features. It would be interesting to use our benchmark to study the impact of data augmentations and other dataset curation techniques (e.g., filtering). We will include the above discussion in Section 4.1.
>
> >Q4: Any insight on WIT differs from LAION
>
> Good suggestion. We think the performance difference between WIT and LAION datasets could be attributed to two main factors:
> First, dissimilarity in data sources used for training - Common Crawl for LAION and unknown data sources for OpenAI's training set. As pointed out by [e], Common Crawl may be a noisier data source (weaker connection between images and associated text) or contain less diverse images.
>
> Second, filtering process in the LAION dataset curation: LAION employed a small-scale ViT-B/32 model for filtering, which may result in a substantial amount of poorly matching image-text pairs. Moreover, recent work [f] suggests that more appropriate curations on LAION give the competitive performance of CLIP models compared to those trained on WIT.
>
> The above further highlights the importance of dataset curation for learning robust CLIP models. We will include the above discussion.
>
> [e] LAION-5B: An open large-scale dataset for training next-generation image-text models. NeurIPS'22
>
> [f] DataComp: In search of the next generation of multimodal datasets, Arxiv 2023
>
> >Q5. Other self-supervised models like MAE … demystify the factors of datasets and objectives
>
> Thanks for this valuable suggestion. During rebuttal, we evaluated three MAE models on three aspects in Figure R-3 in the uploaded PDF.
>
> Here are our observations: 1) Visual factor-level robustness: three MAE models lie on the area of models pre-trained on more data;
> 2) OOD detection: MAE models also lie in the area with other models pre-trained on more data. Some zero-shot CLIP models achieve higher performance than MAE models;
> 3) Calibration: before TS, MAE models have higher uncertainty estimation performance than CLIP trained on LAION while lower than CLIP trained on WIT. Post TS, CLIP models become better than MAE.
>
> >Q6: Is it safe to say that when the dataset size is similar, the supervised objective is still better in calibration, robustness, and OOD detection
>
> Interesting point. According to the discussion in [g], when the dataset size is small (e.g., 15M), CLIP models’ classification accuracy on IN-1K would drop. In contrast, under the same dataset size, supervised objectives achieve high-accuracy models. When considering a similar training set scale, we anticipate the following phenomena:
>
> 1) Factor-level Robustness: CLIP exhibits lower accuracy on each factor while maintaining the same relative robustness trend as models trained on larger datasets.
>
> 2) OOD Detection: CLIP's performance weakens compared to supervised models.
>
> 3) Calibration: Given the impact of training set quantity on CLIP, we expect it to achieve less effective calibration results than supervised models.
>
> [g] Quality Not Quantity: On the Interaction between Dataset Design and Robustness of CLIP

---

> > ### Comment · Reviewer_A6cY · 2023-08-17
> >
> > Thank you for your effort on the thorough response. I think the additional experiments on ImageNet help clarify the coupled objective and training data factors. The answers also mitigate my other concerns. Therefore, I would like to keep my original rating.

---

> > > ### Author Response · Authors · 2023-08-17
> > > **Thank you**
> > >
> > > Dear Reviewer A6cY,
> > >
> > > Thank you for your valuable and constructive suggestions! We are happy to hear that your concerns have been addressed.
> > >
> > > Kind Regards,
> > >
> > > Authors

---

### Official Review · Reviewer_jPAm · 2023-06-30

**Soundness:** 3 good
**Presentation:** 3 good
**Contribution:** 2 fair
**Rating:** 4
**Confidence:** 4

**Summary:**

This paper studies and compares CLIP and CLIP-FT to standard models on a range of different tasks including OOD robustness, OOD detection, and model calibration. The paper constitutes a meta-analysis across different model architectures / training datasets / training algorithms or loss functions.

The authors claim:

####  Robustness
CLIP sometimes outperforms other models on certain visual factor variations, but sometimes underperforms them.
CLIP models are more shape biased (authors claim due to VLM training), resolution increase decreases shape bias.
Fine-tuning makes CLIP behavior more similar to other image models.

####  OOD Detection
CLIP generally performs better than other models in OOD detection; the relationship between IID accuracy and OOD detection performance largely follows accuracy on the line.
Fine-tuning negatively affects performance.

####  Model Calibration
CLIP isn’t significantly more calibrated than other models, data distribution affects calibration.
Temperature scaling makes CLIP more calibrated than other models, and removes dependence on data distribution.
Temperature scaling is apparently more important for CLIP.
When temperature scaled CLIP is better OOD calibrated than other models.

####  Test Time Prompts
Using more prompts generally helps across tasks, except in the case of some visual factor variations where it doesn’t really make a difference

**Strengths:**

- The introduction is well written, motivates the paper and gives a clear overview of the claims.
- Well written, easy to follow
- Objective tone/analysis and solid experimental design
- Some nice points/findings dispersed throughout the paper

**Weaknesses:**

- No clear central argument or claim, paper seems more like a pastiche of different experiments rather than a focused analysis. There is little interpretation of the experimental findings. I have written very detailed questions considering the OOD detection results, but similar questions apply to all sections.
- It looks to me that the paper tries to do too much in one paper and ends up being imprecise / too shallow on interpreting the results. For example, [2] focuses solely on how the data distribution affects robustness. Here, the authors observe this finding and only note that “The above observations highlight the importance of the choice of training source in determining not only the overall accuracy but also the factor-level behaviors of CLIP models. This suggests that visual factor-level robustness should be considered when designing the training source for CLIP models.” This is a “political” answer which does not provide concrete action items for future researchers.
Some claims/analyses are probably wrong:

    *  “Temperature scaling reveals a consistent trend of CLIP models, and they still lie on a distinct trend from other models.” This doesn’t seem to be the case ALL of the time. SSL models look pretty similar to CLIP on the “NLL (Temp-scaled)” and “ECE (Temp-scaled)” Imagenet-A plots. Amount of difference seems to be dataset specific. In fact, I only see a significant effect in ImageNet-A, ECE vs ECE-temp scaled, where the temperature scaling seems to affect all models.
    *   ““This observation indicates that unlike robustness and out-of-distribution detection, the calibration of CLIP models is influenced by both training data distribution and quantity.” I don’t see how this follows from your data - aren’t robustness and OOD detection also influenced by data distribution?
    *  The CLIP training dataset likely overlaps with the datasets used for OOD detection which makes it unclear how meaningful the claims are in regards to CLIP’s good OOD detection performance. This issue should at least be discussed.
    *   OOD detection evaluations are missing important baselines (see details below).

- The paper lacks novelty: The influence of training data / fine-tuning affecting CLIP performance have been studied in detail before [1,2,3]. This paper evaluates many different models and is thus a meta-study of the previous findings. In that case, the literature review should be expanded, and the paper must be better positioned, e.g. in “in [3], the authors investigate the influence of fine-tuning on OOD robustness. We here investigate whether their claims hold across a broader range of models'' or something like this. Though [3] already provides a thorough and careful empirical investigation across many different models, and I am not sure whether this paper offers much beyond the results presented in [3].
- The graphs were sometimes a bit hard to read/interpret - there were a lot of them and usually the conclusion was different/unique depending on the plot.
- Some kind of unifying principle might be nice.

### Minor:
- Please be more specific about results/claims in the abstract.
- 3.1 - Would be nice to explain/list model choices at some point, i.e. refer to Supplement A.2. here
- 3.2 (Robustness) - more explanation of why you chose 10 of the 16 would be nice (robustness)
- Line 40, style / grammar: “our study further study .. “
- Line 42: “training distributions”, it should be distribution
- Line 50, remove the extra space before the full stop
- Line 280: “highlighting their potential for the robust and reliable applications” → “highlighting their potential for robust and reliable applications.

**Questions:**

What is the central claim/argument?
*  Maybe pivot towards study similar to [2] in the context of OOD detection and model calibration? That is: Be more specific on what causes the observed OOD detection results.
*  What about the data distribution matters for OOD detection and model calibration?
In general, I would appreciate some deeper insights, I don’t feel like I learned all that much from this paper.

### Robustness results (Section 4)

Lines 138-144: The authors observe that CLIP models perform better on certain visual factors and worse on others. Some interpretation of the results is necessary to make the observations valuable to the research community. For example, the authors find that fine-tuning on ImageNet changes the robustness properties of CLIP models, but offer no explanation / interpretation of the results.


###  OOD detection results (Section 5):

Line 209: “Upon closer examination of the training distribution, we have observed that the correlation trend between ID accuracy and OOD detection performance is largely dependent on the training source.” -> It is not clear to me how sensible the OOD detection task is for CLIP models. While LAION may not contain e.g. all of the ImageNet training images, doing zero-shot inference is possible because CLIP has seen similar images during training, thus, “zero-shot” is ultimately a misleading term which becomes an issue once OOD detection is considered. It is not clear to me that the datasets used for OOD detection here, namely iNaturalist [52], SUN [53], PLACES [54], TEXTURE [55] and ImageNet-O [7], are actually not part of LAION in the first place, making measuring performance on those datasets an ID task, rather than an OOD task. Thus, I find it unsurprising that the training data distribution matters a lot for this particular task. Further, comparing the performance of models trained on ImageNet which has no intersection with ImageNet-O to models trained on LAION which may have a high intersection with ImageNet-O seems an unfair comparison to me. Could the authors please comment on this issue, in particular, on the sensibility of doing OOD detection for CLIP models trained on e.g. LAION?

Continuing on this point: Line 212: “Moreover, with same ID accuracy, CLIP models trained on WIT exhibit superior OOD detection performance compared to their counterparts trained on LAION on three OOD scenarios. This further indicates the importance of training source selection for CLIP. When developing dataset curation methods, it is valuable to investigate the influence of training sources on OOD detection performance.” I would like to see a more nuanced and detailed discussion on this finding. Is it maybe that WIT just has a larger overlap with the OOD detection datasets? What is the concrete suggestion here?

Line 219: “Some CLIP-FT models even achieve worse OOD detection performance than Zero-shot CLIP models.” This is in line with the points above and what we may be seeing here is that with fine-tuning, the model weights move towards being compatible with ImageNet-1K and further away from the original training distribution which likely overlaps with OOD detection test sets.

Due to all of the points raised above, I find the sentence in Conclusion, line 303: “Furthermore,
while maintaining comparable accuracy on in-distribution dataset, CLIP models tend to exhibit higher performance in OOD detection.” to be misleading.

The recent paper [4] (published after the NeurIPS submission deadline) makes the very similar observation that the CLIP training dataset has a large influence on zero-shot OOD detection, and also remarks on the effects of the fine-tuning procedure. Comparing how the conclusions made in this paper align with theirs would be very helpful. An evaluation on their dataset would also be interesting to see, since they show that Places, Texture and ImageNet-O have severe issues with ID data.
They seem to contradict the notion that CLIP zero-shot is well suited for OOD detection. I see no reason not to compare fine-tuned CLIP models with those fine-tuned on ImageNet 21K, which are very commonly used.
Further, the evaluation of OOD detectors should include strong baseline models like the Mahalanobis detector, compared to which the CLIP based models clearly fall behind according to the evaluations in [4].
Not including the mentioned important baselines (which are also included in [5]) might be the main explanations why the evaluations in Figure 3 (and in the previous papers claiming zero-shot CLIP with MCM being a strong OOD detector) suggest such a positive image of the CLIP based models.


More centralized and focused claims/analyses would raise my score.


#### References:

* [1] Quality Not Quantity: On the Interaction between Dataset Design and Robustness of CLIP
* [2] Data Determines Distributional Robustness in Contrastive Language Image Pre-training (CLIP)
* [3] The Evolution of Out-of-Distribution Robustness Throughout Fine-Tuning
* [4] Bitterwolf et al. ICML 2023: “In or Out? Fixing ImageNet Out-of-Distribution Detection Evaluation”
* [5] Yang et al. NeurIPS 2022: “OpenOOD: Benchmarking Generalized Out-of-Distribution Detection”

**Limitations:**

Limitations were not addressed.

---

> ### Author Rebuttal · Authors · 2023-08-09
>
> >Q1: No clear central argument ... unifying principles
>
> Thanks. Our central argument is the necessity of a comprehensive evaluation of CLIP's robustness. In contrast to current approaches focused on classification accuracy, we propose integrating three new safety-driven objectives: factor-level robustness, OOD detection, and uncertainty calibration. This enables a thorough assessment of critical factors (e.g., training source and prompt) on CLIP's behaviours. Our analysis highlights the significance of training sources on the three objectives.
>
> >Q2-1: Interpretation on robustness … fine-tuning changes robustness
>
> Thanks. As pointed out by Reviewer A6cY (Q1), CLIP uses color-jitter, which could improve its robustness to color and brightness but hurt Pose. Also, fine-tuning CLIP with contrastive loss maintains shape-bias in prediction.
>
> Moreover, as discussed in [a], standard fine-tuning introduces spurious correlation for CLIP and thus hurts robustness.
>
> [a] Masked Images Are Counterfactual Samples for Robust Fine-tuning
>
> >Q2-2: Maybe pivot towards [2] for OOD detection and calibration
>
> Thanks. Following [2], we indeed investigated crucial factors for CLIP, including training source, dataset quantity, test-time prompt, contrastive loss, and architecture. Through this analysis, we emphasize the significance of data distribution: 1) data distribution affects the accuracy trend of zero-shot CLIP in OOD detection and calibration; 2) CLIP's calibration is further influenced by dataset quantity. Additionally, we stress the role of prompt learning, as different prompts can alter the performance trend on both tasks.
>
> >Q3-1: Some claims probably wrong: 'TS reveals a consistent trend of CLIP … distinct from other models'
>
> We clarify that Fig.4 gives two observations: 1) Before TS, CLIP models from different sources and subsets exhibit distinct trends; 2) Post TS, CLIP models exhibit a similar trend, diverging from other models: with comparable classification accuracy, they demonstrate lower calibration errors.
>
> >Q3-2: 'Unlike robustness and OOD detection, calibration is influenced by both training distribution and quantity'
>
> We clarify that CLIP model's factor-level robustness and OOD detection are affected by training distribution, while calibration is influenced by both training distribution and quantity.
>
> >Q4: Lack novelty … training data/fine-tuning influence have been studied in [1-3] … a meta-study.
>
> Our work significantly differs from [1-3], which concentrates solely on classification robustness under distribution shifts. In contrast, our study investigates CLIP models' robustness from three new perspectives: factor-level robustness, OOD detection, and calibration. Furthermore, [1-3] do not analyze how training data and fine-tuning impact CLIP models across these aspects. We give such analysis and comprehensively examine training distribution, quantity, test-time prompt, and fine-tuning schemes, revealing novel observations into CLIP models' behaviour.
>
> >Thanks for raising the insightful discussion on CLIP's sensibility for OOD detection. We think potential data overlap does not compromise CLIP's promising results and will not introduce sensibility. Please see the below:
>
> >Q5-1:  CLIP may have seen similar images during training … “zero-shot” is misleading
>
> We follow the notion of “zero-shot” OOD detection (Ming et, al. 2022): CLIP allows users to redefine ID/OOD classes flexibly without requiring detector retraining.
>
> >Q5-2:  Not clear OOD datasets are not part of LAION ... unfair comparison
>
> We hold the same opinion with the authors of LAION: ‘we do not consider potential test set overlap to be a serious threat for the validity of results’.
>
> Dataset overlap may arise if OOD datasets are also included in Common Crawl. In classification, OpenAI found only a few examples of substantial performance differences due to data overlap. Similarly, as discussed in [4] for OOD, the overlap between IN-21K with NINCO does not cause substantially different changes between models with and without pretraining on IN-21K. This further indicates potential data overlap will not introduce sensibility for CLIP.
>
> >Q5-3: Maybe WIT has a larger overlap than LAION with OOD datasets
>
> The performance gap in OOD (and classification) could be attributed to training source quality and filtering process. Please see Q2/Reviewer A6cY.
>
> >Q5-4: Fine-tuning shifts weights towards IN-1K and away from original distribution that likely overlaps with OOD datasets
>
> Fine-tuning CLIP on IN-IK could learn unsuitable features for OOD detection due to spurious correlation [a]. Further, using a broader set of data/classes from IN-21K may potentially mitigate this issue.
>
> >Thanks for sharing post-submission work [4]. The discussion has strengthened the OOD detection aspect of our study:
>
> >Q6-1: [4] observes CLIP training dataset has a large effect on zero-shot OOD detection and remarks on fine-tuning effect ... fine-tuned CLIP on IN-21K
>
> After careful check, [4] only included two zero-shot CLIP models and did not study training set impact.
>
> IN-12K is a subset of IN-21K with excluded classes having few samples, resulting in approximately 85% overlap [4]. IN-12K is widely used for CLIP fine-tuning. Moreover, [4] reports fine-tuning on IN-21K helps OOD detection, and we observe fine-tuning CLIP on IN-12K is helpful. We will cite [4] and discuss the mutual finding.
>
> >Q6-2: Evaluation on their dataset ... Mahalanobis detector
>
> Thanks. We noted that [4] reports "it is difficult for many OOD detectors to improve consistently over MSP", supporting the rationale for using MSP in our study. During rebuttal, we established our benchmark on NINCO as per [4], using Mahalanobis and MSP. Please refer to Figure R-4 in the uploaded PDF.
>
> Our findings in Sec. 5 align consistently with NINCO results. Further, we observe promising zero-shot CLIP detection accuracy,  compared to other ImageNet models with Mahalanobis detector.

---

> > ### Comment · Reviewer_jPAm · 2023-08-14
> > **Response to the rebuttal**
> >
> > Dear authors,
> > I have read the rebuttal but I find that most of my questions have not been addressed.
> >
> > 1) No central argument in the paper.
> > Your response was: “In contrast to current approaches focused on classification accuracy, we propose integrating three new safety-driven objectives: factor-level robustness, OOD detection, and uncertainty calibration.” The objectives are not new at all, and studying all of them separately does not provide a unifying objective for the storyline. As stated in my original review, the paper reads as a conglomeration of different experiments without much interpretation of the results. In principle, you could have added more safety-driven objectives, such as e.g. mechanistic interpretability, fairness analyses, adversarial attacks etc., to have even more experimental results without a unifying principle. I think the ideas in this paper are still too scattered for it to be accepted at NeurIPS.
> >
> > 2) “Interpretation on robustness … fine-tuning changes robustness” Here, I stated that the results are very similar to Ref [3] (The Evolution of Out-of-Distribution Robustness Throughout Fine-Tuning). I also wrote that the interpretation / analysis of the results is very shallow. The authors responded writing about the effect of Color Jitter in CLIP which does not answer nor concern any of my questions.
> >
> > 3)	“TS reveals a consistent trend of CLIP … distinct from other models”. First, it is impossible to compare the metrics with and without temperature scaling because the y-axes are different. The authors claim that CLIP models do not follow the trends of “other models” and that temperature scaling leads to consistent trends in CLIP models. My issues: 1) I find the first statement to not be true because I do not see any trends in “other models” which CLIP models may follow, e.g. ImageNet-S / ImageNet-V2-A (ECE temp scaled). All models are scattered across the plot and I do not see any distinct trends. 2) Models denoted by different icons do cluster together on some datasets, with and without temperature scaling. But in those cases, all model classes seem to be in somewhat different clusters, and I do not see a notable distinction of CLIP vs other models.
> >
> > 4) Lack of novelty. Yes, the authors here study model performance across several axes in contrast to previous work which examined model behavior carefully along one of those axes. But there is still very little analysis and interpretation of the results. What new insights do we get? As stated in my original review, the insights offered by the authors are not actionable or concrete, such as “The above observations highlight the importance of the choice of training source in determining not only the overall accuracy but also the factor-level behaviors of CLIP models. This suggests that visual factor-level robustness should be considered when designing the training source for CLIP models.” The observation that the training data source influences robustness is trivial and well-known.
> > - As stated in my original review, there are papers which analyze one of the aspects studied here in great detail. For example, [2] (Data Determines Distributional Robustness in Contrastive Language Image Pre-training (CLIP)) analyzes the role of the data distribution. The authors here seem to replicate some of the results of [2], e.g. that the training distribution matters. But then, what is the benefit of the current study if it just reproduces some results of more focused studies? Wouldn’t I be better off reading multiple studies which carefully study one aspect of generalization instead of this paper? I think a meta-study needs to have a lot more analysis and interpretation to add value.
> > 5)	“We hold the same opinion with the authors of LAION: ‘we do not consider potential test set overlap to be a serious threat for the validity of results’. Please provide a citation for this. Were the LAION authors specifically concerned with OOD detection in that statement? Further, there are issues when evaluating OOD robustness of foundation models, please see Ref [4].
> > 6)	OOD findings on NINCO: Contributions: “With comparable in-distribution accuracy, CLIP models are competitive or better in detecting OOD data than other ImageNet models.” I do not see this effect in the attached Figure R-4. The best results on AUROC are achieved by the ImageNet-21K models (orange circles).
> > a.	“Our findings in Sec. 5 align consistently with NINCO results. “ Please be more specific: which  NINCO results align with which of your Sec. 5 results? I find this not to be true. The results in Sec. 5 are very different from the NINCO results, see my argument below.
> > b.	“Further, we observe promising zero-shot CLIP detection accuracy, compared to other ImageNet models with Mahalanobis detector.” I do not share this observation. The best models in this plot are ImageNet-21K models, no?

---

> > > ### Comment · Reviewer_jPAm · 2023-08-14
> > > **Response to author rebuttal (part 2)**
> > >
> > > 7)	Comparing the results in Fig. 3 of the main paper to the NINCO results, there is actually a huge difference. In Figure 3, the CLIP models do seem to be much better on the OOD detection task than all other non-CLIP models. But this is not true on NINCO, where ImageNet-21K models are better. This is significant because the NINCO authors write that “We find that most of the currently used test OOD datasets, including datasets from the open set recognition (OSR) literature, have severe issues: In some cases more than 50% of the dataset contains objects belonging to one of the ID classes. These erroneous samples heavily distort the evaluation of OOD detectors. As a solution, we introduce with NINCO a novel test OOD dataset, each sample checked to be ID free, which with its fine-grained range of OOD classes allows for a detailed analysis of an OOD detector's strengths and failure modes, particularly when paired with a number of synthetic "OOD unit-tests". -> The observation from Figure R-4 that ImageNet-21K models are in fact better at the OOD detection task, is significant and quite different from the observation in Fig. 3.
> > >
> > > I would like to continue the discussion with the authors since my concerns have not been resolved by their rebuttal.

---

> > > > ### Author Response · Authors · 2023-08-14
> > > > **Follow-Up Discussion (2/2)**
> > > >
> > > > > Q5: Discussion on data overlap
> > > >
> > > > Data overlap presents potential concerns for both OOD robustness and detection.
> > > >
> > > > In terms of classification robustness, [c] discusses that data overlap may arise if OOD datasets are also included in Common Crawl (database of LAION). Also, OpenAI found only a few examples of substantial performance differences due to data overlap. The authors of LAION [c] (Page 9, Section6) mentioned that ‘we do not consider potential test set overlap to be a serious threat for the validity of results’.
> > > >
> > > > As for OOD detection, we share the same opinion. An illustrative instance is: as discussed in Ref [4] for OOD, the class data overlap between pretraining dataset IN-21K and OOD test set NINCO does not cause substantially different changes between models with and without pretraining on IN-21K.
> > > >
> > > > Moreover, another OOD detection benchmark unit-test in Ref [4] shows that CLIP remains effective on entirely new and synthetic OOD datasets.
> > > >
> > > > Based on these considerations, we posit that potential data overlap will not introduce significant sensibility for CLIP on OOD detection. We also think it would interesting for future works to better understand the precise effects of data overlap on both OOD detection and robustness. We will include the above discussion in the revision.
> > > >
> > > > [c] LAION-5B: An open large-scale dataset for training next generation image-text models
> > > >
> > > >
> > > > > Q6-1: Which NINCO results align with which of your Sec. 5 results
> > > >
> > > > The results on NINCO align with the observation in Sec. 5:
> > > > 1) For CLIP models from the same source, their ID accuracy correlate with OOD detection (L199);
> > > > 2) Training source influences the trend of CLIP (L209);
> > > > 3) Fine-tuning procedure influences the OOD detection ability of CLIP (Line 217).
> > > >
> > > > > Q6-2: “Promising zero-shot CLIP detection accuracy, compared with models with Mahalanobis detector” … The best models in this plot are IN-21K models
> > > >
> > > > We clarify that with “comparable classification accuracy (e.g, ~80% accuracy)”, zero-shot CLIP models achieve competitive OOD detection accuracy with other ImageNet models.
> > > >
> > > > Second, We did not claim zero-shot CLIP models are better than IN-21K pretrained models. Models pretrained on IN-21K have higher accuracy (>85%) than zero-shot CLIP models. Yet, If we look at the CLIP’s performance trend, a future zero-shot CLIP with higher classification accuracy has potential to achieve promising OOD detection accuracy with those pretrained on IN-21K. Our observation is more conservative and restricted to OOD performance trend/tendency instead of absolute performance comparison.
> > > >
> > > > > Q7: CLIP models do seem to be much better on the OOD detection task than all other non-CLIP models. But this is not true on NINCO, where ImageNet-21K models are better.
> > > >
> > > > Our study did not claim zero-shot models achieve much better results than IN-21K models nor zero-shot CLIP is the best for OOD. Instead, we focus on the impact of fine-tuning, training distribution and quantity on CLIP models. The results on NINCO share the same observation on these factors (Q6-1).
> > > >
> > > > We would like to discuss the mentioned observation in absolute performance comparison between zero-shot CLIP and IN-21k pretrained models as follows,
> > > >
> > > > First, as post-submission work Ref [4] points out that OOD dataset could contain ID classes, which distorts the evaluation of OOD detectors. We concur this may partially explain the difference in performance comparison.
> > > >
> > > > Second, for OOD performance comparison, we think it should also consider the used OOD dataset:
> > > >
> > > > 1) For NINCO, it combine many OOD datasets/ classes together and use the average AUROC for comparison. Combined dataset vs. Single dataset would give different rankings in both models and OOD detectors.
> > > >
> > > > 2) When using another benchmark unit-test in Ref [4], we would have a different observation: IN-21K pre-trained models are not always the most effective.
> > > >
> > > >
> > > > Thank you for bringing up the above discussions. We hope our explanations have addressed your concerns and we remain open to further discussion.
> > > >
> > > > Kind regards,
> > > >
> > > > Authors

---

> > > > > ### Comment · Reviewer_jPAm · 2023-08-15
> > > > > **Discussion with authors**
> > > > >
> > > > > Dear authors,
> > > > > Thank you for your fast response.
> > > > >
> > > > > “We do not study them separately but consistently investigate the impact of crucial factors on CLIP’s behaviour on each objective, including model architecture, training data distribution, training set quantity, fine-tuning, contrastive loss and test time prompt.”
> > > > >
> > > > > - Please point me to the result where the architecture influence is studied for OOD detection of CLIP models? Stating that you analyzed the model architecture on each objective means, there need to be results for CLIP training for ResNet models / ViTs / MLP Mixer models? You write that you use CLIP models with different architectures, but I do not see a plot showing the influence of the architecture?
> > > > > - Considering the “contrastive loss” objective, I believe you do not study this in the paper. You write that you use the MoCo and SimCLR models in the comparisons, where both the training distribution and the objective are changed. This analysis has also been done in [2], and in [A], in a lot of detail.
> > > > > [A] Santurkar et al. “Is a Caption Worth a Thousand Images? A Controlled Study for Representation Learning”
> > > > >
> > > > > **TS-results**: I understand that the two sets of columns show different quantities. But you do compare the results before TS scaling and after TS scaling and thus, ECE (before / after) and NLL (before / after) need to have the same y-axes to be comparable.
> > > > >
> > > > > “Before TS, all zero-shot CLIP models from different sources and subsets do not have a unified trend. Post TS, all zero-shot CLIP models exhibit a similar trend.” I am sorry, but I do not see this. Do you mean with “trend” that they are clustered in a similar region? Under trend, I would understand that you could better fit a linear model to those points, but I don’t think this would work better post TS?
> > > > >
> > > > > **Novelty**: “Following [2], we investigated crucial factors for CLIP: 1) training source, 2) dataset quantity, 3) test-time prompt, 4) contrastive loss, 5) fine-tuning, and 6) architecture.” I disagree with this statement. There is no visible distinction in the plots considering the dataset size or architecture. The authors using models trained on datasets of different sizes and using models with different backbones does not mean their influence was investigated if those factors cannot be distinguised in the plots. For some of the investigated metrics, such as calibration, the influence of dataset size has been studied, but not for the others.
> > > > >
> > > > > **OOD detection results**:
> > > > >
> > > > > “Our study did not claim zero-shot models achieve much better results than IN-21K models nor zero-shot CLIP is the best for OOD. Instead, we focus on the impact of fine-tuning, training distribution and quantity on CLIP models.” -> The third contribution of the submitted paper is explicitly “With comparable in-distribution accuracy, CLIP models are competitive or better in detecting OOD data than other ImageNet models.” Thus, the statement above is explicitly made as part of the contributions. Other researchers might cite the paper as evidence for this claim.
> > > > >
> > > > > I still do not see the point substantiated that CLIP models are competitive with other ImageNet models in terms of OOD detection. Particularly for standard models which are fine-tuned from IN-21K, no CLIP based models seem to achieve similar performance. Since such models are very widely used and comparable to (or rather even cheaper than) CLIP in terms of training compute, being competitive with those is what a reader would expect when reading the stated contribution claim. The argument that they are not comparable in the 80% accuracy regime since the analysis does not include IN-21K model with this accuracy to me just indicates that no statement about this can be derived here. See also the conclusion of the next point.
> > > > >
> > > > > "We noted that [4] reports "it is difficult for many OOD detectors to improve consistently over MSP", supporting the rationale for using MSP in our study." -> [4] explicitly observe that “most feature-based methods, like Mahalanobis […] outperform the MSP-baseline by a clear margin”. With this in mind a fair comparison between zero-shot CLIP models (which use feature-based MCM) and IN-1K classifiers should also use a feature based method for the latter. Using MCM for all models (which according to [4] Table 3 works well for fine-tuned CLIP) could also be interesting besides Mahalanobis in order to have a direct comparison with the same method.
> > > > > The observation in the main paper that “Some CLIP-FT models even achieve worse OOD detection performance than Zero-shot CLIP models” could likely simply be explained by the employed OOD detection method.
> > > > > Particularly regarding the Mahalanobis values, the potential for “future zero-shot CLIP with higher classification accuracy” to be competitive with IN-21K pre-trained models is speculative and not apparent.
> > > > >
> > > > > Best,
> > > > > Reviewer jPAm

---

> > > > > > ### Author Response · Authors · 2023-08-16
> > > > > > **Follow-up discussion**
> > > > > >
> > > > > > Dear Reviewer jPAm,
> > > > > >
> > > > > > Thank you for your helpful feedback and suggestions. Please see the clarification below,
> > > > > >
> > > > > > > No visible distinction in the plots considering the dataset size or architecture
> > > > > >
> > > > > > We follow the analytical methodology of seminal work [a] (so as a series of following works like Ref [1-3]) to study the influential factor. In the context of an overarching performance trend exhibited by all models, any factor that leads the associated models to deviate from this trend is identified as influential. In our study, we mainly highlight the influential factors (e.g., data quantity for calibration and test-time prompts for calibration and OOD detection).
> > > > > >
> > > > > > Based on the suggestion, we can divide the figures into several subfigures to show the effect of each factor. Or including a legend to illustrate each factor. We will also release our database and plots code, allowing users to check and add factors.
> > > > > >
> > > > > > [a] Measuring robustness to natural distribution shifts in image classification.
> > > > > >
> > > > > > > Model Architecture OOD detection of CLIP models
> > > > > >
> > > > > > As illustrated in Sec. 3.1 and Supp., our study covers different CLIP models with various architectures. The architecture does not break the performance trend: CLIP models lie in two trends determined by the training source only. Namely, it is not the key factor for performance tendency. A legend will be included to denote the architecture.
> > > > > >
> > > > > > > Contrastive loss
> > > > > >
> > > > > > We indeed follow Sec. 7 (Effect of contrastive training losses) of Ref [2] to study the contrastive pre-training models (e.g., SimCLR and SimSiam).
> > > > > >
> > > > > > During the rebuttal, we additionally studied the impact of contrastive loss for fine-tuning on shape-texture bias.
> > > > > >
> > > > > > > Trend in TS-results
> > > > > >
> > > > > > Thanks. First, we can use the same scale for y-axes. Second, the trend does not mean it should be linear. Based on the suggestion, we will use the following text:
> > > > > >
> > > > > > For post-TS, 1) when using NLL as the metric, CLIP models exhibit a clear correlation between their NLL error and classification accuracy. 2) Using ECE as the metric, CLIP models have the similar relationship on ImageNet-A; On IN-Val/V2-A/S, they cluster in a similar region and tend to have lower calibration errors than most of other ImageNet models. We will update Lines 246-273 to make it clear.
> > > > > >
> > > > > >
> > > > > > > About OOD results.
> > > > > >
> > > > > > Thanks for sharing your feedback and suggestions based on the post-submission benchmark NINCO.
> > > > > >
> > > > > > First, we strictly follow the protocol in [c] to report CLIPs' performance on commonly used benchmarks (e.g., iNaturalist and Texture). The main findings (L199, L209, and L217) are obtained on these benchmarks.
> > > > > >
> > > > > > On these commonly used benchmarks, the observation is obtained: "with comparable in-distribution accuracy, CLIP models are (tend to be, L207) competitive or better in detecting OOD data than other ImageNet models". We did not claim it is our contribution but one of key observations obtained from our study.
> > > > > >
> > > > > > It seems the reviewer raises question on this observation using post-submission dataset NINCO. We respectfully note that we did not claim this observation is universal or hold on any future datasets like NINICO. We can specify the scope of the commonly used benchmark to make it more concrete.
> > > > > >
> > > > > > [c] Delving into Out-of-Distribution Detection with Vision-Language Representations.
> > > > > >
> > > > > > About the post-submission Ref [4]: on new benchmark NINCO, the authors benchmark the results using various OOD detectors and models.
> > > > > >
> > > > > > 1) Considering this, the authors point out "it turns out to be difficult for many OOD detectors to improve consistently over the baseline of MSP" (Sec.1, Page2). We think it is reasonable to use MSP when comparing various models in our study. Following reviewer’s comment, we also reported the results using Mahalanobis on NINCO;
> > > > > >
> > > > > > 2) The observation that “most feature-based methods, like Mahalanobis […] outperform the MSP-baseline by a clear margin” is made for *ViT-B only* (Sec. 4.1. Results on NINCO). When using other models (e.g., ResNet-50), Mahalanobis could achieve low detection performance (Table 3 in Ref [4]);
> > > > > >
> > > > > > 3) On NINCO, the impact of fine-tuning, training distribution and quantity on CLIP models (L199, L209, and L217) remain consistent with our main paper. Our observation is more conservative and restricted to OOD performance trend/tendency. With the observed trend, we think it is reasonable to say promising OOD detection results can be achieved by future zero-shot CLIP with higher classification accuracy. We note that this trend-based analytical methodology is core of a series of works in CLIP (e.g., Ref [1-3]).
> > > > > >
> > > > > > 4) As discussed in Ref [4], Mahalanobis may not be suitable for zero-shot CLIP models; Ref [4] opts for MCM over Mahalanobis while investigating such models. Further, Ref [4] did not use MCM for conventional classifiers. Instead, Ref [4] newly proposed a new variant called RCos based on the notion of MCM. We will discuss the results using RCos besides Mahalanobis for other models on NINCO.
> > > > > >
> > > > > > Best regards,
> > > > > >
> > > > > > Authors

---

> > > > > > > ### Comment · Reviewer_jPAm · 2023-08-18
> > > > > > > **Discussion with the authors**
> > > > > > >
> > > > > > > Dear authors,
> > > > > > >
> > > > > > > considering the TS results, your new statement sounds ambiguous. "For post-TS, 1) when using NLL as the metric, CLIP models exhibit a clear correlation between their NLL error and classification accuracy." -> All models exhibit a clear correlation between the NLL metric and classification accuracy both before and post TS, not just CLIP models. You should write "With and without TS, when using NLL as the metric, all considered models exhibit a clear correlation between their NLL error and classification accuracy.
> > > > > > >
> > > > > > > When using ECE as metric, it might be that the drop in the drop in the ECE metric is stronger for CLIP models compared to other models, but this is currently impossible to judge because the y-axes are different. It also really depends which models have been used and compared to.
> > > > > > >
> > > > > > > **OOD detection results**: "On these commonly used benchmarks, the observation is obtained: "with comparable in-distribution accuracy, CLIP models are (tend to be, L207) competitive or better in detecting OOD data than other ImageNet models". **We did not claim it is our contribution but one of key observations obtained from our study.**" This is not correct. This precise sentence is the third contribution of the paper, thus, this has been claimed as a contribution. Given the new results on NINCO, please remove this contribution because it does not hold. Other researchers might cite this paper for this contribution and it would be wrong. Stating that future zero-shot models might obtain promising OOD detection results in the future is a conjecture and cannot be used to back up this contribution.
> > > > > > >
> > > > > > > Best,
> > > > > > > Reviewer jPAm

---

> > > > > > > > ### Author Response · Authors · 2023-08-19
> > > > > > > >
> > > > > > > > Dear Reviewer jPAm,
> > > > > > > >
> > > > > > > > Thank you for your helpful comments. The text will be revised in line with your recommendations:
> > > > > > > >
> > > > > > > > > Illustration of TS results
> > > > > > > >
> > > > > > > > Thank you for your nice suggestion. We will align the y-axes of Fig. 4 to the same scale for pre/post-TS comparison when using the same metric. Additionally, we will revise lines 266-270 as per your recommended phrasing to enhance the clarity of our observations.
> > > > > > > >
> > > > > > > > > OOD detection results: Given NINCO, please remove "with comparable in-distribution accuracy, CLIP models are (tend to be, L207) competitive or better ..." researchers might cite this and it would be wrong
> > > > > > > >
> > > > > > > > Thank you for your meticulous feedback. We will carefully revise the corresponding text (L60, L207, and L304) to focus *solely* on the impact of fine-tuning, training distribution, and quantity on CLIP's OOD detection capabilities. Additionally, we will include and discuss the results on the post-submission dataset NINCO. We will also point out that the impact of fine-tuning, training distribution and quantity on CLIP models (L199, L209, and L217) remain consistent on NINCO.
> > > > > > > >
> > > > > > > > Thank you again for your helpful feedback.  Please let us know if you have any other questions.
> > > > > > > >
> > > > > > > > Kind Regards,
> > > > > > > >
> > > > > > > > Authors

---

> > > ### Author Response · Authors · 2023-08-14
> > > **Follow-Up Discussion (1/2)**
> > >
> > > Dear Reviewer jPAm,
> > >
> > > Thank you for your constructive feedback. Please see the discussion below:
> > >
> > > > Q1: Further discussion on central argument
> > >
> > > We call for attention on safety-related objectives beyond classification accuracy alone, when evaluating CLIP’s robustness.
> > >
> > > Recent findings highlight the pivotal role of training distribution on CLIP’s classification robustness, whereas other factors exhibit limited influence. However, this paper raises the concern that relying solely on training set distribution does not suffice for ensuring complete robustness.
> > >
> > > We study the three representative objectives (visual-factor level robustness, OOD detection, and calibration). Note that, we do not study them separately but consistently investigate the impact of crucial factors on CLIP’s behaviour on each objective. Following analytical approach outlined in Ref [2], we consider model architecture, training distribution, training set quantity, fine-tuning, contrastive loss and test time prompt. Our experiments emphasize factors like test-time prompts (OOD and calibration) and training set quantity (calibration) remain important considerations for a comprehensive evaluation of CLIP's robustness.
> > >
> > > Last, we view our work as a starting point to call for attention on safety-related objectives. It would be interesting to include other objectives (e.g., the mentioned fairness).
> > >
> > > > Q2: “Interpretation on robustness … fine-tuning changes robustness” … results are very similar to Ref [3]
> > >
> > > First, Ref [3] studies the effect of fine-tuning on effective robustness in overall classification accuracy. In contrast, we consider the visual factor-level robustness as well as texture-shape bias. Ref [3] did not report the observations on this perspective. We will cite [3] and discuss the difference.
> > >
> > > Second, we observe fine-tuning CLIP could help some visual factors (e.g., Pattern)  but hurt others (e.g., Texture). We speculate standard fine-tuning introduces spurious correlation [a]. This may lead to a bias for CLIP towards specific visual properties, thereby compromising factor-level robustness on some factors. Further, fine-tuning CLIP with contrastive loss maintains shape-bias in prediction (Q1/ Reviewer A6cY).
> > >
> > > Moreover, as suggested by Reviewer A6cY (Q3), data augmentations used in zero-shot CLIP could improve robustness to related factors, but with spill-over effects to unrelated factors.
> > >
> > > [a] Masked Images Are Counterfactual Samples for Robust Fine-tuning
> > >
> > > > Q3: “TS reveals a consistent trend of CLIP … distinct from other models”
> > >
> > > First, Y-axes are ECE (first two columns) and NLL (last two columns) of Fig. 4. We use “Temp-scaled” to denote TS is used for calibration.
> > >
> > > Second, based on feedback, we restate the observations of Fig. 4 to make it clear:
> > > 1) Before TS, all zero-shot CLIP models from different sources and subsets do not have a unified trend;
> > > 2) Post TS, all zero-shot CLIP models exhibit a similar trend. Also, other models do not follow this trend of CLIP. We did not claim other models have a trend instead they are scattered as reviewer mentioned;
> > > 3) Post TS, zero-shot CLIPs lie below most other groups models: with similar classification accuracy, CLIP tends to achieve lower calibration error than other models.
> > >
> > > > Q4: Discussion on Novelty
> > >
> > > We respectfully disagree with the statement that our work is not novel. We contribute a comprehensive study to better understand CLIP's robustness.
> > >
> > > [New perspectives] Following analytical approach in Ref [2], we study crucial factors for CLIP: 1) training source, 2) dataset quantity, 3) test-time prompt, 4) contrastive loss, 5) fine-tuning, and 6) architecture. Through extensive analysis, we underscore the data source's critical role across three new perspectives while also uncovering overlooked factors, such as data quantity's impact on calibration. To our knowledge, we are the first to study the impact of these factors on the three new perspectives.
> > >
> > > [New observations] Unlike prior works (e.g., Ref [1-3])  focusing on overall classification, we further explore new objectives and obtain more insights.  For instance, high overall classification robustness does not guarantee robustness to individual visual factors and the shape bias in CLIP predictions.
> > >
> > > Our investigation also uncovers under-explored aspects. Key observations include the reduction of shape bias in CLIP predictions after ImageNet fine-tuning. Contrary to previous observations, CLIP models are not consistently more calibrated than other ImageNet models, owing to training data distribution and quantity. The impact of training source and fine-tuning strategies is evident in their impact on OOD detection performance.
> > >
> > > [Benchmark Application] In line with recent study [b] that highlights the importance of dataset curation in CLIP, our benchmark provides comprehensive metrics to full assess the curated datasets, alongside classification accuracy.
> > >
> > > [b] DataComp: In search of the next generation of multimodal datasets

---

### Official Review · Reviewer_rieU · 2023-07-01

**Soundness:** 3 good
**Presentation:** 3 good
**Contribution:** 3 good
**Rating:** 5
**Confidence:** 3

**Summary:**

Authors closely study the robustness of vision-language models. They try to investigate their robustness in terms of common visual attributes, detecting OOD inputs, and their power in providing calibrated predictions. They consider many different CLIP models and other vision encoders with different architectures and training procedures to have a comprehensive study and fairly compare CLIP models with other ones.

They provide some more detailed findings about these models w.r.t to the aforementioned criteria.

**Strengths:**

+ This paper runs an extensive set of experiments using various models, various datasets, and under different settings.
+ Therefore, these results will be insightful for practical use cases where people want to decide which model to use or diagnose possible errors/failures of their models on different conditions.


**Weaknesses:**

+ I didn't see enough new ideas in this paper.
+ I mean, running extensive studies is definitely valuable, practical, and insightful, but is there other similar work published in NeuriPS where scaling up and running more experiments is the main contribution? I would appreciate it if the authors correct me in understanding their main contribution and change the rating correspondingly.


**Questions:**


Are these results generalizable and reliable? i.e., are we going to see similar trends for future CLIP models as well? I would want to get some more insights about these findings. At this point, this study looks completely empirical and I am not convinced if they are statistically significant. Can we really trust those plots and argue some general statements?

**Limitations:**

Limitations are addressed.

---

> ### Author Rebuttal · Authors · 2023-08-09
>
> >Q1: Did not see enough new ideas ... running extensive studies is definitely valuable, practical, and insightful, but is there other similar work published in NeurIPS where scaling up and running more experiments is the main contribution?
>
> We appreciate your recognition of our extensive studies and experiments. We clarify that our contributions go far beyond the scale of experiments:
>
> - New perspectives: In contrast to existing analysis paradigms (e.g., [a,b,c]) centred around overall classification accuracy, we advocate for the integration of three novel safety-driven objectives: factor-level robustness, OOD detection, and uncertainty calibration. This approach allows us to thoroughly assess and understand the impact of critical factors on CLIP models’ robustness, including training source, quantity, network structure, test-time prompt, and fine-tuning strategy.
>
> - New observations: Our extensive investigation into CLIP models uncovers several previously under-explored aspects. Key observations include the reduction of shape bias in CLIP predictions following ImageNet fine-tuning. Contrary to previous assumptions, CLIP models are not consistently more calibrated than other ImageNet models, owing to training data distribution and quantity. The significance of training sources and fine-tuning strategies is evident in their impact on OOD detection performance. Furthermore, while test-time prompts do not affect CLIP's visual factor-level robustness, they influence the trends in OOD detection and uncertainty calibration.
>
> - Comprehensive data curation metric: Aligned with recent research [d] underscoring the importance of training dataset curation, our benchmark introduces comprehensive metrics to evaluate curated datasets alongside their classification performance.
>
>     [a] Quality Not Quantity: On the Interaction between Dataset Design and Robustness of CLIP, NeurIPS'22
>
>     [b] Data Determines Distributional Robustness in Contrastive Language Image Pre-training (CLIP), ICML'22
>
>     [c] The Evolution of Out-of-Distribution Robustness Throughout Fine-Tuning, TMLR'22
>
>     [d] DataComp: In search of the next generation of multimodal datasets, Arxiv 2023
>
> > Q2. Are these results generalizable and reliable? ... similar trends for future CLIP models as well? Can we really trust those plots and argue some general statements?
>
> Thanks for raising this discussion. **First**, to ensure the validity and reliability of our study, we followed established practices [a,b,c] meticulously, giving careful consideration to various factors such as training sources, network architectures, fine-tuning procedures, test datasets, and other comparable ImageNet models.
>
> We adopt methodologies from previous research to study each objective: ImageNet-X [e] for analyzing visual factor-level robustness; Cue-conflict stimuli [f] to examine shape bias in model predictions; [g] to gauge CLIP's zero-shot OOD detection capabilities; and [h] to delve into the quality of uncertainty estimation.
>
> [e] ImageNet-X: Understanding Model Mistakes with Factor of Variation Annotations, ICLR'23
>
> [f] ImageNet-trained CNNs are biased towards texture; increasing shape bias improves accuracy and robustness, ICLR'19
>
> [g] Delving into Out-of-Distribution Detection with Vision-Language Representations, NeurIPS'22
>
> [h] Revisiting the Calibration of Modern Neural Networks, NeurIPS'21
>
> **Second**, we firmly believe that the insights gained from our analysis extend beyond the CLIP models evaluated in this study and are applicable to future models as well. To validate this point, during the rebuttal, we expanded our investigation to include the very latest CLIP models (beyond the submission deadline). These models are trained on DataComp or CommonPool [d]: they are trained on the same training source with LAION but with different dataset quantities.
>
> The results shown in Figure R-3 in the uploaded PDF have demonstrated that our observations hold true for these newly included models, further affirming the scalability and generalizability of our findings.
>
> **Third**, to facilitate further research and analysis on CLIP models, we will release our experimental setups, including the database and plot codes.
>
> The above discussion will be included in the revised version.

---

> > ### Comment · Reviewer_rieU · 2023-08-17
> >
> > Dear authors, thanks for your response. i increased my score.

---

> > > ### Author Response · Authors · 2023-08-17
> > > **Thank you**
> > >
> > > Dear Reviewer rieU,
> > >
> > > Thank you for your positive assessment and helpful suggestions on our work.
> > >
> > > Best,
> > >
> > > Authors

---

### Official Review · Reviewer_9D3z · 2023-07-06

**Soundness:** 4 excellent
**Presentation:** 4 excellent
**Contribution:** 4 excellent
**Rating:** 7
**Confidence:** 3

**Summary:**

This paper analyzes the CLIP model's robustness through a large number of experiments, including three main points: resilience to visual factor variations, calibrated uncertainty estimations, and the ability to detect anomalous inputs.

**Strengths:**

1.  The experiments in this paper are very sufficient, the research content is solid, and some new views are proposed from the experimental results. There is a rich analysis of the robustness of CLIP.
2. This paper makes a significant contribution to the field by providing a comprehensive evaluation of CLIP models.  Furthermore, the experimental findings presented in this paper offer valuable insights for future endeavors aiming to enhance the out-of-distribution (OOD) detection performance and robustness of CLIP models.

**Weaknesses:**

There is no deeper analysis of the reasons behind these experimental results in this paper.

**Questions:**

1. Please provide some details about the experiments.
2. I want to ask whether parameter-efficient fine-tuning (PEFT) methods, such as LORA or Adapter, hurt the performance of OOD detection, given that fine-tuning sometimes can.

**Limitations:**

The authors has provided the limitations of their work.  This paper has done a lot of experiments and provided some novel discoveries that I think contribute to the CLIP community.

---

> ### Author Rebuttal · Authors · 2023-08-09
>
> >Q1. There is no deeper analysis of the reasons behind these experimental results in this paper
>
> This work emphasizes the incorporation of three new safety-driven objectives: factor-level robustness, OOD detection, and uncertainty calibration. This enables a comprehensive assessment of critical factors on CLIP models' robustness, encompassing training source, quantity, network structure, test-time prompt, and fine-tuning strategy. Our benchmark highlights the significance of training sources in this context.
>
> Building on the insightful comments from all reviewers, during the rebuttal, we delved deeper into several intriguing aspects, including the performance difference between LAION and WIT (Q4/Reviewer A6cY), the retention of shape bias using contrastive loss for fine-tuning (Q1/Reviewer A6cY), the possibility of spill-over effects from data augmentation impacting unrelated robustness factors (Q3/Reviewer A6cY), and potential data overlaps in OOD detection (Q5/Reviewer jPAm).
>
> We acknowledge the potential for deeper analysis of our observations and consider the above discussion as a starting point that could inspire further research.
>
> >Q2. Please provide some details about the experiments
>
> In the appendix, we provide illustrations of the publicly available models and datasets used in our study. We follow: ImageNet-X [a] to conduct analysis on models' visual factor level robustness; Cue-conflict stimuli [b] to study shape-bias in model decisions; [c] to understand the CLIPs' performance on zero-shot OOD detection and [d] to investigate the quality of uncertainty estimation.
>
> [a] ImageNet-X: Understanding Model Mistakes with Factor of Variation Annotations, ICLR'23
>
> [b] ImageNet-trained CNNs are biased towards texture; increasing shape bias improves accuracy and robustness, ICLR'19
>
> [c] Delving into Out-of-Distribution Detection with Vision-Language Representations, NeurIPS'22
>
> [d] Revisiting the Calibration of Modern Neural Networks, NeurIPS'21
>
> Furthermore, we are dedicated to facilitating future research on CLIP analysis, and as part of this commitment, we will release the plot codes and evaluation codes for the community to use and build upon.
>
> >Q3. I want to ask whether parameter-efficient fine-tuning (PEFT) methods, such as LORA or Adapter, hurt the performance of OOD detection, given that fine-tuning sometimes can
>
> Insightful point. During the rebuttal, we studied the impact of parameter-efficient fine-tuning methods on OOD detection. In Figure R-2 of the uploaded PDF, we report the results of 8 CLIP models fine-tuned by CoOp[e] and Tip-Adapter[f]. We find that both two methods increase the classification accuracy of CLIP models, while decreasing the OOD detection performance. It would be interesting to further study the effect of PEFT methods on OOD detection. Also, increasing both classification and OOD detection performance would be a promising direction.
>
> [e] Learning to Prompt for Vision-Language Models, IJCV'22
>
> [f] Tip-Adapter: Training-free Adaption of CLIP for Few-shot Classification, ECCV'22

---

> > ### Comment · Area_Chair_Choe · 2023-08-18
> >
> > Dear Reviewer 9D3z
> >
> > Could you please kindly check the other reviews and rebuttal, and raise your concerns if you have any? We are already close to the end of the author-reviewer discussion phase.
> >
> > Thanks,
> >
> > Regards, AC

---

### Official Review · Reviewer_8Bsp · 2023-07-11

**Soundness:** 4 excellent
**Presentation:** 3 good
**Contribution:** 3 good
**Rating:** 6
**Confidence:** 4

**Summary:**

This paper aims to provide a comprehensive evaluation of robustness for pretrained vision-language models. Specifically, the authors benchmark around 100 pretrained models/classifiers. Based on these empirical results, this paper also provides corresponding discussions and analysis.

**Strengths:**

1. The robustness problem of CLIP like model is a valuable topic to study.
2. This paper provides a very comprehensive benchmark for CLIP robustness problem. It may contribute to several benefit for the following research.
3. Corresponding discussion and analysis are solid to further inspire study in this area.

**Weaknesses:**

Even if I am still concerning the technical contribution of this paper for the NeurIPS conference, I recognize the workload and benchmarking work of this paper. Thus, only a little comments for the weaknesses:
1) What about further adding image-text retrieval evaluation and analysis? since CLIP can be used for both classification and retrieval. It may make this paper more solid.
2) I would like to discuss with other reviewers about the contribution significance of this benchmarking work to adjust my final score.

**Questions:**

Please see weaknesses.

**Limitations:**

Limitations have been discussed by the authors.

---

> ### Author Rebuttal · Authors · 2023-08-09
>
> >Q1. Further adding image-text retrieval evaluation and analysis ... make this paper more solid
>
> Thanks. Following this Insightful suggestion, we included retrieval tasks on MS-COCO during the rebuttal. Figure R-1 in the updated PDF reports the results. We plot retrieval performance (image-to-text and text-to-image retrieval) relative to zero-shot image classification accuracy on the ImageNet validation set.
>
> On both retrieval tasks, we observe that retrieval performance correlates with image classification performance. In addition, CLIP models trained on different training sources appear to have different trends. We will include and discuss this new analysis in the revised version.
>
> >Q2. Contribution significance of this benchmarking work
>
> Thanks for raising this discussion. Beyond the existing analysis of CLIP's classification robustness, our study advocates for the integration of three novel safety-driven objectives: factor-level robustness, OOD detection, and uncertainty calibration. This allows us to thoroughly assess and understand the impact of critical factors on CLIP models’ robustness, including training source, quantity, network structure, test-time prompt, and fine-tuning strategy. Through extensive and careful analysis, our benchmark underscores the impact of training sources across three new objectives.
>
> Furthermore, our extensive studies have unveiled several previously unknown aspects of CLIP models, deepening our understanding of their robustness behaviours. For instance, we demonstrate the shape bias in CLIP models' predictions, which diminishes after fine-tuning on ImageNet. Contrary to existing findings, CLIP models are not always more calibrated than other ImageNet models, and we attribute this to the impact of both training data distribution and quantity. Also, training sources and fine-tuning procedures have crucial effects on their  OOD detection performance. Furthermore, test-time prompts do not impact CLIP's visual factor-level robustness but influence the performance trend of OOD detection and uncertainty calibration.
>
> Moreover, as very recent research [a] highlights the importance of training dataset curation, our benchmark provides comprehensive metrics to assess the curated datasets, alongside the classification performance.
>
> [a] DataComp: In search of the next generation of multimodal datasets, Arxiv 2023

---

> > ### Comment · Area_Chair_Choe · 2023-08-18
> > **discussion**
> >
> > Dear Reviewer 8Bsp
> >
> > Could you please kindly check the other reviews and rebuttal, and raise your concerns if you have any. We are already close to the end of the author-reviewer discussion phase.
> >
> > Thanks,
> >
> > Regards,
> > AC

---

### Author Rebuttal · Authors · 2023-08-09

Dear Reviewers,

Thank you for your detailed and thoughtful feedback. Inspired by your valuable suggestions, we have added more experimental analyses and included the suggested discussions.

We summarize the experiments in the uploaded PDF:
- In Fig. R-1, we evaluate the retrieval performance of zero-shot CLIPs. We observe retrieval performance of CLIPs correlates with their classification performance. Also, the performance trend is influenced by training dataset distribution.

- In Fig. R-2, we study the OOD performance of 8 new CLIP models finetuned by parameter efficient fine-tuning methods: CoOp and Tip-Adapter. Both methods improve models’ classification performance but lead to performance drops in OOD detection.

- Fig. R-3 includes post-submission CLIP models and three MAE pre-trained models. We observe they are aligned with our observations in the main paper.

- In Fig. R-4, we expand the OOD benchmark on NINCO, using both MSP and Mahalanobis detectors. Our findings in Section 5 of the main paper align consistently with NINCO results. Also, zero-shot CLIP shows promising accuracy, compared to other ImageNet models with Mahalanobis detector.

- Moreover, we show the retention of shape bias using contrastive loss for fine-tuning (Q1/Reviewer A6cY)

Last, we greatly appreciate all reviewers recognize our study is comprehensive, sufficient, and solid. We emphasize that our main contributions go beyond just the scale of experiments. Specifically, our study unveils three novel perspectives (factor-level robustness, OOD detection, and uncertainty calibration) that contribute to a comprehensive understanding of CLIP models' robustness.

We hope our response has addressed the initial concerns. Please let us know if you have any other questions.

Kind Regards,

Authors

---

### Decision · Program_Chairs · 2023-09-21

**Decision:**

Accept (poster)

**Comment:**

This paper undertakes a comprehensive study on the robustness of multiple CLIP models, analyzing their performance with respect to various visual factors, out-of-distribution detection, and calibrated uncertainty estimations. All reviewers agree that this paper is well-written, and the empirical experiments are well-designed and comprehensive. Meanwhile, some concerns are raised: 1) the technique novelty is not very significant, as some conclusions can already be reasonably drawn from existing works, and 2) some conclusions are confusing and probably wrong.

The rebuttal phase was constructive, with the authors engaging proactively to clarify and fix multiple issues in this paper. Given four reviewers agreed to accept it and one reviewer voted for borderline rejection (but be okay for accepting it), the final decision is accept.

For the final version of the manuscript, the authors must integrate all the experimental data and clarifications discussed during the rebuttal phase, particularly the exchanges with Reviewer jPAm, to ensure the high standards of NeurIPS publications.